# Antioxidant Systems as Modulators of Ferroptosis: Focus on Transcription Factors

**DOI:** 10.3390/antiox13030298

**Published:** 2024-02-28

**Authors:** Carolina Punziano, Silvia Trombetti, Elena Cesaro, Michela Grosso, Raffaella Faraonio

**Affiliations:** 1Department of Molecular Medicine and Medical Biotechnology, University of Naples Federico II, 80131 Naples, Italy; carolina.punziano@unina.it (C.P.); silvia.trombetti@unina.it (S.T.); elena.cesaro2@unina.it (E.C.); 2Department of Veterinary Medicine and Animal Productions, University of Naples Federico II, 80137 Naples, Italy

**Keywords:** ferroptosis, redox systems, transcription factors, oxidative stress, lipid peroxidation

## Abstract

Ferroptosis is a type of programmed cell death that differs from apoptosis, autophagy, and necrosis and is related to several physio-pathological processes, including tumorigenesis, neurodegeneration, senescence, blood diseases, kidney disorders, and ischemia–reperfusion injuries. Ferroptosis is linked to iron accumulation, eliciting dysfunction of antioxidant systems, which favor the production of lipid peroxides, cell membrane damage, and ultimately, cell death. Thus, signaling pathways evoking ferroptosis are strongly associated with those protecting cells against iron excess and/or lipid-derived ROS. Here, we discuss the interaction between the metabolic pathways of ferroptosis and antioxidant systems, with a particular focus on transcription factors implicated in the regulation of ferroptosis, either as triggers of lipid peroxidation or as ferroptosis antioxidant defense pathways.

## 1. Introduction

In about ten years of research investigations, ferroptosis emerged as a type of cell death with a pivotal role in preventing or even facilitating the progression of many diseases (reviewed in [1,2,3]). In particular, ferroptosis is now considered a good therapeutic option to treat multiple forms of cancer and arrest tumor growth [3,4,5], as well as a causal factor in neurodegenerative disorders [6], senescence/aging [7], different blood diseases [8], and kidney- and ischemia-reperfusion injuries [9,10].

Ferroptosis has been recognized as a type of regulated, iron-dependent cell death with distinct morphologic features and molecular mechanisms differing from other forms of regulated dying pathways like apoptosis, necrosis, and autophagy-induced cell death [11]. In fact, ferroptosis is a programmed cell death that occurs when lipid peroxides accumulate because of a loss of redox homeostasis; thus, the control of lipid peroxidation represents the main regulator of ferroptosis (reviewed in [12]). Metabolic pathways triggering increased redox-reactive molecules like iron, reactive oxygen species (ROS)/reactive nitrogen species (RNS), enzymatic lipid peroxidation, and failure of endogenous antioxidant systems, mainly glutathione peroxidase 4 (GPX4)/glutathione (GSH) and coenzyme Q, can fuel ferroptosis [13]. Although the molecular mechanisms and/or metabolic pathways are still poorly understood, ferroptosis seem to be under the control of numerous transcription factors [14,15], with most of them acting as transcriptional modulators of genes implicated in lipid peroxidation and redox homeostasis, as well as iron levels.

Aim: To highlight the relationships among oxidative stress, regulatory pathways, and ferroptosis.

## 2. Ferroptosis as a Biological Program: Features and General Mechanisms

Beginning with the identification of apoptosis, the archetypal example of “programmed cell death,” many non-apoptotic cell death pathways, including several regulated necrotic pathways such as necroptosis, disulfidptosis, netosis, entosis, and crupoptosis, have been identified in recent years [16,17]. Although ROS and RNS are crucial in most types of cell death, it is only now becoming clear how they regulate and characterize the specific type of cell death that takes place in any given cell [18,19]. Indeed, since cellular signaling and stress response are ROS-dependent processes, it is not surprising that cell death pathways are dependent on the type and quantity of ROS/RNS production, which can function as a rheostat, enabling the activation of various cell death pathways and possibly cross-talks with other forms of cell death. As a type of non-apoptotic cell death, ferroptosis is a ROS/RNS-dependent cell death caused by the imbalance of three metabolic pillars: thiols, polyunsaturated phospholipids, and iron [20].

### 2.1. Role of Iron in Ferroptosis

Even though iron is necessary for cell division and survival, its accumulation combined with an increase in lipid-derived radicals and lipid peroxides may cause cell death [11]. Notably, in this context, the catalytic activity of enzymes involved in both ROS formation (lipoxygenases/LOXs, cytochrome P450/CYPs, xanthine oxidase, NADPH oxidases, mitochondrial complex I and III) and decomposition (catalase, peroxidases) relies on loosely bound iron or iron-complexes (i.e., heme or [Fe-S] clusters).

Three pools of iron are most likely representative of the iron species thought to be directly involved in ferroptosis: (i) the first one includes catalytic centers of non-heme iron proteins like LOXs utilizing iron at the active site; (ii) the storage iron pool represented by ferritin, and (iii) the major fraction represented by the redox-active, non-coordinated Fe^2+^ present in the cytosolic labile iron pool (LIP) that can be transported into mitochondria and utilized for heme and iron-sulfur cluster synthesis (Figure 1). Lysosomes have a very large LIP because of their role in recycling endogenous iron sources like mitochondria and ferritin and in absorbing exogenous iron. Interestingly, it has recently been reported that ferroptosis in cancer cells cannot occur unless functional lysosomes are present [21,22]. 

In addition, intracellular iron overload can be related to dysregulated levels of several proteins involved in iron metabolism/homeostasis including transferrin, the carrier of blood ferric iron (Fe^3+^); transferrin receptor 1 (TFR1), which is the primary iron importer into the endosomal compartment; DMT1 (divalent metal transporter 1), which releases iron into the cytoplasmatic iron-labile pool after it has been reduced to ferrous iron Fe^2+^ by ferrireductases (e.g., STEAP3); ZIP14, mediating non-transferrin-bound iron import; ferritin, which stores iron; and the iron exporter ferroportin (FPN1) [23,24]. Also, autophagic degradation of ferritin (ferritinophagy), the intracellular LIP, and heme degradation by the inducible enzyme heme oxygenase 1 (HO-1) generates iron, in addition to carbon monoxide (CO) and biliverdin [25,26,27,28]. 

Notably, autophagy is basically a recycling pathway that can lead to cell death when it becomes disturbed [29]. An example of iron transport protein dysregulation that results in iron overload is seen in ferroptosis-sensitive Ras-mutant cells, which have elevated TFR1 and decreased ferritin levels [30]. Notably, during ferroptosis, the genes related to iron metabolism are mostly upregulated and numerous transcription factors are implicated in such control (reviewed in [31,32]).

Under physiological conditions, the ferrous iron of the LIP pool is kinetically unstable, therefore extremely reactive, and can catalyze Fenton reactions, which decompose endogenously produced hydrogen peroxide (H_2_O_2_) into highly reactive intermediate species, like hydroxyl and peroxyl radicals (HO^•^, HOO^•^) or high-valence oxo-ferryl species [33]:Fe^2+^ + H_2_O_2_ → Fe^3+^+ HO^•^ + OH^−^
Fe^3+^ + H_2_O_2_ → Fe^2+^+ HOO^•^ + H^+^
high-valence oxo-ferryl species

Several cellular components, particularly phospholipids containing polyunsaturated fatty acid chains (PUFAs), are expected to be attacked and oxidatively damaged by the Fenton-derived ROS species, which will eventually cause lipid hydroperoxides or other hydroperoxides to become active, causing damage to membranes and other intracellular structures, thereby triggering cell death processes [34]. Additionally, it is possible that the iron centers of LOX enzymes catalyze the production of hydroperoxyl lipids (LOOH), the primary enzymatic products of lipid peroxidation, while Fe^2+^ from the LIP takes part in the secondary reactions of LOOH decomposition to create electrophilic products of lipid peroxidation that are oxidatively truncated [20,35,36].

### 2.2. Iron Accumulation and Lipid Peroxidation

The induction of ferroptosis employes specific pathways involving redox active iron and/or iron-dependent peroxidation enzymes. Iron is a fundamental cofactor for several enzymes involved in oxidation–reduction reactions due to its ability to exist in two ionic forms: ferrous (Fe^2+^) and ferric (Fe^3+^) iron. As mentioned before, in the ferroptosis context, the enzymes that catalyze lipid peroxidation are mostly heme or non-heme iron enzymes [37,38]. Consequently, iron homeostasis plays a crucial role in lipid peroxidation.

Lipids are essential components of cell membranes that maintain the structure and control the function of cells. The cellular lipid composition and the related mechanisms by which the cell imports, synthesizes, stores, and catabolizes different lipids crucially shape ferroptosis sensitivity [39,40]. In more detail, the type of unsaturated phospholipids in the cell membrane determines how cells are sensitive to ferroptosis, with PUFAs increasing the sensitivity of cells to ferroptosis whereas monounsaturated fatty acids (MUFAs) have inhibitory effects toward this cell death process [39,41].

PUFAs are a family of lipids with two or more double bounds that can be classified in omega-3 (n-3) and omega-6 (n-6) fatty acids (FAs), according to the location of the final double bond relative to the molecule’s terminal methyl terminus. The principal n-6 FAs are linoleic acid (C18:2), arachidonic acid (AA, C20:4), and its elongation product, adrenic acid (AdA, C22:4). PUFAs are important substrates for unsaturated phospholipid synthesis and also for the generation of lipid peroxides, mostly when they are incorporated into phosphatidylethanolamines (PE) [42]. The activation and esterification of AA and AdA into PE is carried out by specific enzymes, namely the acyl-coenzyme A (CoA) synthetase long-chain family member 4 (ACSL4) and lysophosphatidylcholine acyltransferase 3 (LPCAT3), rendering the ACSL4/LPCAT3 axis the principal pro-ferroptotic system that under iron overload conditions can push lipid nonenzymatic autoxidation reactions or eventually increase lipoxygenase activities (as discussed later in detail).

Lipid peroxidation (Figure 2) relies on the reaction of oxygen-derived free radicals with PUFAs, resulting in lipid peroxyl radicals and hydroperoxides. The conversion of PUFAs incorporated into membrane phospholipids to peroxide PUFAs represents the initiation step to drive ferroptotic cell death [12]. Once formed, lipid peroxides can interact with ferrous iron to generate peroxyl radicals that can then abstract hydrogens from neighboring acyl chains in the lipid membrane environment to propagate the lipid peroxidation process. As mentioned before, specific PUFAs like arachidonic acid and adrenic acid are peroxidized to drive ferroptosis; however, other PUFA-containing phospholipids, including various diacyl and ether-linked phospholipids, are oxidized during ferroptosis and likely contribute to this process in a context-dependent manner [43]. In summary, the membrane phospholipid composition and therefore the endogenous pool of PUFAs dictates ferroptosis sensitivity.

In any case, although there is still much debate over whether the lipid peroxidation process is strictly controlled by enzyme activities or if it starts as uncontrollable free radical reactions, the generation of toxic lipid-derived peroxide species involves different forms of catalytically active iron. In order to better comprehend the molecular signature of ferroptosis, it is crucial to critically assess the precise role that iron plays during this process, as well as to promote the development of anti-ferroptotic strategies based on iron chelation [44].

Normally, lipid peroxides are maintained at physiological levels by sophisticated intracellular antioxidant systems (e.g., GPX4/GSH, FSP1/CoQ_10_, GCH1/BH_4_), which are generally inhibited or down-regulated during ferroptosis [45].

### 2.3. Lipid Peroxidation by Non-Enzymatic and Enzymatic Reactions

Ferroptosis is uniquely characterized and driven by overwhelming membrane lipid peroxidation, which leads to altered ion fluxes and ultimately to plasma membrane permeabilization [12]. Overall, the situation is quite flexible: a wide range of lipid substrates and starting sources may be able to support the lipid peroxidation events that ultimately lead to ferroptotic cell death. More specifically, the generation of toxic lipid-derived peroxide species may result from non-enzymatic or enzymatic processes, both involving different forms of catalytically active iron. Evidence for the important role of lipid peroxidation in the execution of ferroptosis also comes from inhibition studies. Many potent and specific inhibitors of ferroptosis terminate the lipid peroxidation process by acting as lipophilic radical-trapping antioxidants, thus bolstering the link between lipid membranes and ferroptosis.

The chemistry of lipid peroxidation is well understood and has been extensively reviewed elsewhere [44]. Here, we will briefly mention non-enzymatic or enzymatic processes contributing to the ferroptotic process (a more detailed discussion is given in Section 3). Different phases of initiation, propagation, and termination are involved in the lipid peroxidation mechanism. Anyway, the generation of toxic lipid-derived peroxides encompasses general reaction schemas that are similar in non-enzymatic and enzymatic processes: the oxidation starts with the abstraction of a bis-allylic hydrogen and consequent formation of lipid radical species, and after the addition of molecular oxygen, they become peroxyl radicals and then hydroperoxides. Numerous types of ROS/RNS (see Section 3.1), particularly in the presence of iron, can initiate lipid peroxidation and push ferroptosis. Mitochondria are also sites of substantial ROS generation that may contribute to the initiation of lipid peroxidation, but their role remains controversial [34,46]. Enzymes that can oxidize lipids include members of the lipoxygenase (LOX) and cyclooxygenase (COX) families and oxidoreductase NADPH-cytochrome P450 reductase (POR) isoforms, as well as fatty acyl-CoA reductase1 (FAR1) (as discussed in Section 3.1.3). However, since in ferroptosis, PUFA phospholipids have been found to be peroxidation substrates instead of free PUFA, LOXs appear to be the most likely candidate enzymatic catalysts of ferroptotic peroxidation based on the observation that COXs oxidize free rather than esterified PUFA [20,41]. In addition, it is to be highlighted that enzymatic peroxidation is highly selective for specific substrates and products, in contrast to the random profile observed for free radical peroxidation. Therefore, even though the mechanistic reactive intermediates of these peroxidation reactions may be similar, the fundamental questions of regulation, specificity, and selectivity still need better understanding, since this will have a significant impact on the development of novel pro- and anti-ferroptotic therapeutic approaches [47]. Accordingly, ferroptosis inhibitors such as ferrostatin-1 (Fer-1) and liproxstatin-1, as well as endogenous antioxidants like α-tocopherol, GSH, and N-acetylcysteine (NAC), inhibit lipid peroxidation, thus demonstrating that ferroptosis induction requires lipid peroxidation and iron-dependent ROS.

As a kind of PUFA-hydroperoxide-dependent cell death, ferroptosis can be theoretically initiated/carried out in all biological membranes, including mitochondria, endoplasmic reticulum (ER), peroxisomes, Golgi apparatus, and lysosomes. Previous reports have observed the activation of the ER stress-related unfolded protein response during ferroptosis [48,49]. To identify the membranes of subcellular structures essential for ferroptosis and unveil the possible sequence of peroxidation, A. N. von Krusenstiern et al. performed a deep evaluation of the structure, activity, and distribution of ferroptosis inhibitor/inducers using novel, powerful technologies [43,50]. In that paper, the authors identified ER membranes as the initial and crucial sub-compartment of lipid peroxidation, even if various other subcellular membranes can produce lipid peroxidation. Thus, the authors suggested an ordered progression: initially lipid peroxides accumulate in the ER membrane and later in the plasma membrane; however, it is not clear if the peroxidation can spread to other membranes, or if it occurs independently at different stages and with different rates.

### 2.4. Ferroptosis and Physio-Pathological Processes

Many physio-pathological processes have been correlated with ferroptosis [2,51], including aging and neurodegeneration, senescence, tumorigenesis, blood diseases, kidney disorders, and ischemia–reperfusion injuries (Figure 3).

#### 2.4.1. Ferroptosis in Aging

Aging is a natural life process characterized by a gradual deterioration of physiological mechanisms, resulting in impaired functions of the organs and increased morbidity and mortality due to a combination of environmental and genetic factors. Hallmarks of aging processes include overproduction of ROS species, exhaustion of antioxidant systems, imbalance in oxidation and antioxidant processes [52], and the up-regulation of oxidative stress-induced pro-inflammatory mediators [53,54]. More recently, aging has also emerged as a process related to dysregulated iron homeostasis with intracellular iron stores accumulating with increasing age [55]. Therefore, since the discovery of the ferroptotic mechanisms, a link between aging and ferroptosis has been postulated in a variety of age-related processes, including cancer, neurodegenerative, and cardiovascular diseases [56]. As an example, in this section, we briefly introduce the impact of ferroptosis on these diseases.

#### 2.4.2. Ferroptosis in Cancer Cells

Many links correlate tumors with ferroptosis. Active metabolism in cancer cells is causative of high ROS production [3]. Hallmarks of cancer metabolism also include elevated iron requirements that under oxidative stress make cancer cells more susceptible to ferroptosis as compared to their normal counterparts. In addition, ferroptosis is regulated by numerous tumor-related genes and signaling pathways [57,58]. Several mechanisms involved in ferroptosis, including system Xc-, GPX4, lipid peroxidation, and GSH metabolism, mediate biological processes such as oxidative stress and iron overload and appear to be dysregulated in many cancer types [59,60,61,62].

However, given the extraordinary complexity and heterogeneity of this group of disorders, sensitivity to ferroptosis may vary greatly among different types of cancer cells. For example, p53, which exerts a critical role in inhibiting tumorigenesis and other cancer processes such as invasion, metastasis, and metabolism, has been reported to control ferroptosis-related genes to prevent tumor growth [63,64]. However, as discussed in detail in Section 4.6 it recently emerged that p53 has opposing effects on ferroptosis regulation in different tumor cells. Thus, ferroptosis susceptibility may be modulated by p53 in a tissue-and cell type-specific manner.

More recently, ferroptosis has been proposed as a strategy to overcome drug resistance in cancer cells, which are often resistant to apoptosis and standard therapies [65,66]. However, cancer cells may employ a variety of genetic or epigenetic strategies to combat these metabolic and oxidative stresses [67]. For example, they may up-regulate the antioxidative transcription factors nuclear factor, erythroid derived 2, and like 2 (NRF2) or increase the expression of solute carrier family 7 member 11 (SLC7A11), which reduces the cell’s vulnerability to ferroptosis. Therefore, whether a cancer cell is more sensitive or resistant to ferroptosis induction depends on its specific genetic background. Additionally, through adaptive mechanisms, ferroptosis may contribute to immunity against cancer. Therefore, because ferroptosis can be an immunogenic or immunosuppressive type of cell death, its function in cancer is rather controversial and needs further studies [68].

#### 2.4.3. Ferroptosis in Leukemia

Normal hematopoiesis requires iron, the depletion of which impairs the proliferation and differentiation of hematopoietic cells [69]. Leukemic cells have higher iron concentrations and transferrin expression than other cancer cells and this facilitates the accumulation of iron-derived damage, especially to membrane lipids. Accordingly, a large body of work has recently highlighted the role of ferroptosis in acute myeloid leukemia (AML). Metabolic pathways typically altered in AML and related to ROS/RNS production include enzymes like nitric oxide synthase (NOS), CYP, NAD(P)H oxidase (NOX), and COX (reviewed in [70]). The high frequency of ROS/RNS overproduction in leukemia clearly implies that they are related to the etiology of this disease [71]. Oxidative stress plays a dual role in the development of hematologic malignancies [72]. Chemotherapeutic drugs induce apoptosis by producing high levels of ROS, which suppress tumor growth [73]. Low levels of ROS, on the other hand, protect AML cells from apoptosis and increase treatment resistance, cell migration, growth, and proliferation [61,74]. AML cells are particularly sensitive to iron overload, and although there is strong evidence that transferrin is highly expressed in these cells with increased binding activity, the exact effect of transferrin on AML cells is unclear [70]. Therefore, ferroptosis is emerged as a therapeutic target in leukemia. Indeed, as reviewed by Weber et al. [75], there are four main categories under which current methods of treating AML fall: iron chelators, modification of iron metabolism-related proteins, production of ferroptosis, and administration of antileukemic drugs through ferritin [75]. Of note, all of these categories are based on the modulation of iron metabolism.

#### 2.4.4. Ferroptosis in Heart Failure

Heart failure (HF) is a worldwide epidemic condition that is becoming increasingly common and endangers people’s health. Cardiomyocytes play a key role in maintaining the physiology of the heart and the loss of these caused by cardiovascular disease can speed up the development of HF [76]. During HF, there is an increase in LIP production causing a release of free iron in the myocardium that can alter redox balance and promote ferroptosis in cardiomyocytes [77]. Studies have shown that ferroptosis can also cause a systemic inflammatory response that induces hypertrophy and the death of cardiomyocytes and leads to chronic adverse ventricular remodeling [78,79]. Thus, inhibition of ferroptosis can prevent inflammation and consequently, cardiomyocyte death, while preserving normal cardiac function [80]. A recent investigation revealed that in cardiomyocytes (H9c2 cells) treated with isoprenaline or erastin in a rat HF model induced by aortic banding there was an excess of lipid peroxidation, elevated iron content, and decreased cell viability [81,82]. Also, ferritin plays a key role in maintaining iron homeostasis in the heart. In fact, ferritin H-deficient mice showed reduced expression of SLC7A11 in heart cells, while selective overexpression of SLC7A11 increased GSH levels and prevented cardiac ferroptosis [83]. Research has thus uncovered the regulatory mechanisms and function of ferroptosis in cardiac disease, offering novel treatment options and insights. Nevertheless, not enough is known about ferroptosis’s function in this area to create a successful treatment plan.

#### 2.4.5. Ferroptosis in Neurodegenerative Diseases

Among age-related conditions, neurodegenerative diseases are frequent causes of disability and among the most common causes of death worldwide. The abnormal metabolism of lipids, of which the brain is especially rich, leads to ferroptosis, which can greatly contribute to acute damage to the central nervous system. Indeed, dysregulation of iron homeostasis and excessive ROS in the brain are common hallmarks of neurodegenerative diseases, including Alzheimer’s disease (AD) and Parkinson’s disease (PD) [84]. When neurons die, their iron-containing inclusions are released. This condition generates a vicious circle: in the absence of efficient iron removal systems, iron accumulates in brain areas and promotes the generation of ROS species, causing other neurons to undergo ferroptosis and further reducing the levels of antioxidant defenses [85,86]. Consequently, with iron-dependent oxidative stress serving as a key indicator of cellular ferroptosis, the notion that ferroptosis is inextricably linked to neurodegenerative diseases is receiving growing support. As a consequence, controlling ferroptosis has become a new treatment option for these disorders. In the case of AD, iron chelators delay the disease onset by preventing neuronal death [87,88].

In Parkinson’s disease, the age-related reduced capacity to regulate iron absorption also contributes to iron overload, which is common among older people. Notably, ferroptosis features like iron accumulation, GSH depletion, lipid peroxidation, and an increase in ROS species have been observed in PD. Accordingly, inhibitors of ferroptosis have the potential therapeutic role to protect the nervous system in PD [89,90,91].

## 3. Redox Homeostasis and Antioxidant Systems: Links to Ferroptosis

Cells undergoing ferroptosis fail to maintain redox homeostasis, skewing toward metabolic pathways that fuel lipid peroxidation. There are three major pathways involved in this type of cell death: those increasing the levels of ROS/RNS and lethal lipid peroxides, those engaging in molecular systems that foster the release/accumulation of redox-active iron, and those inactivating antioxidant defenses and lipid repair, albeit the exact cascade/s, if they exist, have not yet been well defined chronologically [51]. The next sections briefly introduce the concept of redox homeostasis and the link between oxidative stress–ROS/RNS–lipid peroxidation and the ferroptotic process.

### 3.1. ROS/RNS and Redox Homeostasis in Ferroptosis

Ferroptosis is a ROS/RNS-associated cell death: perturbation of the redox status provoking the accumulation of ROS/RNS, mainly evoked by iron increase and downstream lipid peroxidation, represent indeed the principal hallmarks of ferroptosis [11,92]. Accordingly, ferroptosis can be counteracted by ROS/RNS scavengers, by enhanced antioxidant activities, as well as by specific compounds such as ferrostatin or deferoxamine, able to trap peroxyl radicals or chelate iron, respectively (reviewed in [93]). ROS/RNS are molecules possessing peculiar biochemical features, and under physiological conditions, their levels are finely regulated. Indeed, endogenous ROS/RNS, when produced in a controlled manner, participate in important proliferative cascades, behaving as crucial signaling molecules [94]. This highlights the need for biological systems to have the levels and quality of ROS/RNS under strict control. The term “redox homeostasis” is used to indicate that cells maintain appropriate levels of ROS/RNS for physiological oxidant cues (known as eustress) [95]. However, cells can experience oxidative challenges (termed distress) due to high ROS/RNS levels that can trigger oxidative damage to the biomolecules. As such, it is important for cells to detect oxidants and manage them to control the risk of damage. In agreement with this, redox homeostasis is linked to specific endogenous systems that detect changes in redox status and engage downstream antioxidant pathways to try to restore the redox balance [38].

#### 3.1.1. ROS/RNS Sources in Ferroptosis

The terms ROS and RNS refer to different molecules with variable reactivity including free radicals and non-radical species (reviewed in [18,94,96,97,98]). The generation of ROS occurs during aerobic metabolism by the mitochondrial electron transport chain (ETC), or from specific enzymatic reactions as well as through the non-enzymatic Fenton reactions that in the presence of free iron or copper directly produce ROS (reviewed in [98,99,100]) (Figure 2). Examples of ROS are the non-radical hydrogen peroxide (H_2_O_2_) and various oxygen-centered radicals like the superoxide anion (O_2_^•−^), the hydroxyl radical (HO^•^), and the hydroperoxyl radical (H/ROO^•^).

H_2_O_2_ is the most relevant non-radical ROS, and it possesses a relatively low reactivity. At specific concentrations, it is the main component of signaling cascades as reviewed elsewhere [94,101]. The principal targets of H_2_O_2_ are the thiol of cysteine residues, thus leading to modifications in protein activities. H_2_O_2_ is produced enzymatically by cytochrome P450s (CYPs), cyclooxygenases (COXs), and various oxidases (XOs) to support signaling cascades; H_2_O_2_ is also generated transiently by the different superoxide dismutase (SOD1-3) isoforms, located both inside and outside the cells. Of note, a recently identified enzyme producing H_2_O_2_ and involved in ferroptosis is the oxidoreductase NADPH-cytochrome P450 reductase (POR), which will be discussed later. It is worth mentioning that H_2_O_2_ molecules can cross membranes directly or through aquaporin channels, therefore initiating membrane damage. H_2_O_2_ participates in lipid peroxidation through reactions that are heavily reliant on the presence of free iron and copper. Specifically, H_2_O_2_ in the presence of Fe^2+^ and Cu^2+^ is the main source of hydroxyl radicals, like HO^•^, HOO^•^, and ROO^•^ (referred to as Fenton reactions), which represent the most reactive ROS species able to trigger non-enzymatic oxidation in a wide range of biological molecules [94,102], including lipids. As herein discussed, the main targets of these types of ROS are PUFAs, especially those present in biological membranes, provoking the generation of phospholipid peroxides (PL-PUFA-OOH), a causal event in ferroptosis.

O_2_^•−^ can originate accidentally in vivo due to the “leakage” of electrons from the mitochondrial ETC as well as from enzymatic reactions involving dedicated enzymes like various NAD(P)H oxidases (NOXs), xanthine oxidase (XO), and uncoupling nitric oxide synthase (u-eNOS).

NOXs include a family of transmembrane enzymes and constitutively provide localized O_2_^•−^ radicals. In particular, NOX4 and dual oxidase 1 and 2 (DUOX1/2), by retaining O_2_^•−^ longer, favor its spontaneous dismutation to H_2_O_2_ [103]. Production of ROS by NOX1-4 has been implicated in the ferroptosis of cancer cells, provoking lipid peroxidation [104,105,106,107]. Dixon et al. demonstrated that selective inhibition of NOX1-4 activity suppressed ferroptosis induced by erastin in Ras-mutated cancer cells [11]. NOXs use NADPH as a reducing cofactor and since NADPH also serves to recycle oxidized glutathione (GSSG) by the glutathione reductase (GR), NOX overexpression can consequently impoverish cells of GSH, thus exacerbating pro-oxidant conditions [108]. In addition, the activity of NOXs in the process of ferroptosis is increased by the dipeptidyl-peptidase-4 (DPP4) protease, which is a p53-regulated protein [104,105]. NOXs also constitute an important downstream source of H_2_O_2_ because of reactions involving SOD enzymes and/or xanthine oxidase, amino acid oxidase, glucose oxidase, and urate oxidase enzymes that directly produce peroxides from a 2-electron reduction of oxygen. However, O_2_^•−^ and H_2_O_2_ are molecules with limited chemical reactivity because, as reported before, they can be enzymatically inactivated by SOD isoenzymes and by catalase, respectively.

ROS produced by mitochondria also seem to play an important role in the ferroptosis cascade (reviewed in [45,46]), albeit this is considerably debated, firstly because mitochondrial ROS mainly trigger apoptosis and other types of cell death. Mitochondrial ROS originate not only from partial reduction of molecular oxygen by ETC but also from the tricarboxylic (TCA) cycle, as well as from iron overload reactions, thus revealing even more diverse sources of mitochondrial ROS. In a pioneering study, Gao et al. demonstrated that mitochondria-depleted cells (through Parkin-mediated mitophagy) were less sensitive to ferroptosis induced by cysteine starvation or by erastin (that inactivates cystine import, as discussed below), and that the pharmacological inhibition of ETC prevented the accumulation of lipid peroxides, regardless of inducers [109]. Of note, these effects cannot be reproduced in ferroptosis induced by RSL3, a direct GPX4 inhibitor, indicating that independently of mitochondria, this enzyme plays a key role in ferroptosis protection [109]. Other sources of mitochondrial ROS in ferroptosis are represented by two interrelated metabolic pathways: glutaminolysis and tricarboxylic cycle (TCA). The enzyme glutaminase 2 (GLS2), a gene target of p53, converts glutamine to glutamate, which is then deaminated by glutamate dehydrogenase (GDH) to α-keto-glutarate (αKG), an intermediate metabolite of the TCA cycle. The increase in αKG can activate TCA as well as fatty acid synthesis, consequently boosting both ROS production and lipid peroxidation [109,110]. In line with this, Suzuki and colleagues recently demonstrated that Gls2-deficient mice or GLS2-depleted human cancer cells are significantly resistant to ferroptosis and this can contribute to the onset of liver cancer. The authors also proved that GLS2 ablation in hepatic cells causes a decrease in lipid peroxides upon treatment with ferroptosis inducers (erastin and RSL3), with a modest effect on GSH content, implying that GLS2 works through αKG-dependent activation of the TCA cycle and consequently ETC increases (the glutaminolysis–TCA–ETC axis) [111].

Also related to mitochondrial ROS–ferroptosis, Shin et al. demonstrated that the dihydrolipoamide dehydrogenase (DLD), a subunit of the TCA α-keto-acid dehydrogenase (KGDH) complex, can push ferroptosis induced by cystine deprivation or erastin [112]. DLD was previously reported to be a source of mitochondrial ROS [113]. Indeed, DLD gene silencing lowers ROS as well as lipid peroxides generated under cystine depletion [112].

In addition to glutaminolysis, decarboxylation of glucose-derived pyruvate can fuel TCA cycle and ETC activity as well as lipid biosynthesis, thereby facilitating ROS generation and lipid peroxidation; a recent work indeed reported that the enzyme pyruvate dehydrogenase kinase 4 (PDK4) confers resistance to ferroptosis by suppressing pyruvate decarboxylation/fatty acids synthesis [114]. ROS production can also result from an increase in mitochondrial membrane potential (ΔΨm), an event that unleashes ferroptosis upon erastin treatment or cystine starvation [45]. Of note, membrane hyperpolarization represents another difference between ferroptosis and apoptosis: indeed, apoptosis is mostly associated with a ΔΨm decrease [115].

Iron being the metal present in a greater quantity in mitochondria [116], it is the main protagonist of ROS/lipid peroxide formation through Fenton reactions. In addition, iron and its derivate molecules, such as heme or iron–sulfur [Fe-S] clusters, are necessary for the activities of ETC complexes. Iron import may be mediated by voltage-dependent anion channels (VDACs), which favor the entry of most metabolites into mitochondria [117]. For example, iron accumulates in mitochondria after erastin treatment, probably because of VDAC 2/3 channel opening [118], therefore leading to iron-dependent ferroptosis. Of note, the discovery of mitochondria-specific defenses against ferroptosis [119,120] suggests that albeit mitochondria metabolic activities can facilitate ferroptosis, these organelles are equipped with efficient anti-ferroptosis systems. For example, an increase of the iron–sulfur protein (2Fe-2S) mitoNEET (also termed CISD1, CDGSH iron sulfur domain 1), can prevent iron and ROS accumulation by mediating the export of iron and sulfur ions from mitochondria, thereby protecting them from ferroptosis [121,122]. Since mitochondria are a site of specific ROS-producing processes, their involvement in ferroptosis is almost obvious, however, when mitochondrial physiological functions are intact, cells are resistant to ferroptosis: indeed, β-oxidation and the formation of Fe/S clusters boost resistance to ferroptosis.

HO^•^ is one of the most reactive ROS. Because of its poor selectivity, it can attack all biomolecules at or near diffusion-mediated rates. This radical can be produced through non-enzymatic reactions between H_2_O_2_ and free iron/copper involving the complex Fenton chemistry. In addition, among the ROS species, HO^•^ might also be produced by the decomposition of peroxynitrite anion (ONOO^−^). Importantly, the hydroxyl radical cannot be eliminated by enzymatic reaction/s, hence it can produce deleterious effects on different cell structures, either directly or indirectly, especially when produced in proximity to membrane lipids [20].

The most relevant RNS comprise the intercellular messenger nitric oxide (NO^•^) generated by different NOS enzymes from the oxidation of L-arginine (the amino acid), the nitrogen dioxide (NO_2_^•^) radical and the peroxynitrite anion, a powerful oxidant [18,98,99,102,123,124] (Figure 2). The NO^•^ molecule rapidly reacts with superoxide anion generating ONOO^−^; therefore, NO^•^ is often considered to be a toxic species, albeit it is also able to terminate lipid peroxidation propagation reactions (as discussed below) [125]. Peroxynitrite can interact irreversibly with different biomolecules (thiols, iron/sulfur centers, zinc fingers, and metalloproteins), finally leading to cytotoxic events that may evoke apoptotic or necrotic cell death [126]. The involvement of ONOO^−^ in lipid peroxidation was first suggested in the atherosclerosis process [127,128] and then reported in other experimental models as well as in a variety of diseases like cardiovascular and neurodegenerative diseases (reviewed in [129]).

#### 3.1.2. Process of Lipid Peroxidation

It is now widely accepted that peroxidation of PUFAs located in membrane phospholipids (PL-PUFAs) represents a crucial driver of ferroptosis, albeit an open question is how PL peroxidation/damage initiates the molecular cascades that evoke cell death signals [51]. Indeed, free fatty acids do not drive this type of cell death; on the contrary, specific PL-PUFA, essential components of all cell membranes, are particularly susceptible to peroxidation [43]. As shown in Figure 2 and Figure 4, there are three different ways by which peroxidation (dioxygenation) of PUFAs can be executed: (1) free radical reactions (a non-enzymatic radical-chain process); (2) photo-oxidation involving singlet oxygen; and (3) enzyme-mediated peroxidation reactions (this route will be discussed in more detail).

(1)The radical-chain process (or radical chain reaction) consists of three sequential non-enzymatic events: initiation, propagation, and termination (Figure 2). Free reactive oxygen-centered radicals, in particular HO^•^ and HOO^•^, mainly originate from Fenton reactions and can initiate the lipid peroxidation process of different types of PL-PUFAs, frequently those present in biological membranes. In fact, by abstracting a hydrogen atom from a methylene carbon inside the acyl chain double bounds, HO^•^ and HOO^•^ leave an unpaired electron on the carbon, generating a reactive carbon-centered lipid radical (PL-PUFA^•^); this last can also originate in a spontaneous manner inside the acyl chain harboring the “C=C” double bond (autooxidation/autocatalytic) [130]. Usually, this lipid radical undergoes molecular rearrangement to form an internal conjugated diene, which by reacting with O_2_ rapidly produces a lipid peroxyl radical (PL-PUFA-OO^•^). These molecules can subsequently remove hydrogen atoms from adjacent PL-PUFA chains or can combine with each other in several ways (propagation event), producing lipid hydroperoxides (PL-PUFA-OOH), especially if they encounter metals like iron or copper, that by Fenton reactions can also push lipid autoxidation [131]. Thus, if not interrupted by chain-breaking antioxidants (a termination step), an internal radical propagation and peroxidation of lipid radical species can result in a large amount of lipid peroxides [132,133]. The comparatively low dissociation energy of O-O bonds causes PL-PUFA-OOH cleavage, which produces a variety of secondary oxidation products, many of which have an oxygen-containing functionality and a shortened carbon chain with strong electrophilic tendencies. The most common electrophiles are aldehydic, oxo-, and epoxy groups, which can be found at various points along the hydrocarbon chain. Either the truncated phospholipid or the remaining shorter PUFA fragment can form an electrophilic group. Furthermore, persistent exposure to iron and/or copper leads to the decomposition of lipid hydroperoxides with the production of harmful carbonyl compounds like unsaturated 4-hydroxynonenal (4-HNE), malondialdehyde (MDA), and acrolein or end products [134] that can further foster destabilization of cell membranes, finally evoking the breakdown of membrane integrity and consequently ferroptosis [135,136].(2)The second mechanism involves the non-radical singlet oxygen (^1^O_2_), originating from O_2_ by light energy transfer and/or from endogenous enzymatic reactions involving COX, LOX, and myeloperoxidase enzymes, or from photosensitizer endogenous agents like bilirubin, porphyrins, flavins, pterins, melanin/melanin precursors, vitamin K, and B6 vitamers that could absorb light and transmit energy to O_2_ producing ^1^O_2_ (Figure 2) [137,138,139,140]. By representing an excited state of molecular oxygen, singlet oxygen is highly electrophilic and can rapidly react with various molecules, including unsaturated lipids, especially those present in cell membranes. The mechanism differs from lipid free radical autoxidation in that the singlet oxygen directly reacts with the double bond producing lipid peroxyl radicals, more similar to the reaction performed by the HOO^•^ radical.(3)Lipid peroxidation by direct enzymatic mechanisms principally involves members of the lipoxygenase (LOX) family coupled to ACSL4/LPCAT3 activity and the oxidoreductase NADPH-cytochrome P450 reductase (POR) isoforms as well as fatty acyl-CoA reductase1 (FAR1), as discussed in the next section.

#### 3.1.3. Enzyme-Mediated Lipid Peroxidation (Figure 2 and Figure 4)

##### LOX Enzymes

In mammals, the oxidoreductase lipoxygenases with dioxygenation activity are mainly implicated in a metabolic cascade termed the “LOX pathway” producing leukotrienes, one of the classes of bioactive lipids known as eicosanoids [141]. The eicosanoid family also includes prostaglandin, thromboxane, and lipoxin mediators, and the term “eicosanoids” identifies molecules originating from 20-carbon PUFAs (“eicosa” meaning 20 in Greek). Specifically, the LOX pathway converts PUFAs, especially arachidonic acid (C20:4, n-6) and linoleic acid (C18:2, n-6), into their corresponding hydroperoxyl derivatives by introducing molecular oxygen at stereospecific positions of the acyl chain. Mechanistically, LOXs, which are non-heme iron-containing enzymes, produce metabolites in a multistep reaction: in the case of arachidonic acid (AA) as a substrate (as shown in Figure 2 and Figure 4), in the first step, AA undergoes a hydrogen abstraction with electron rearrangement, generating Fe^2+^ from Fe^3+^, which is necessary to produce the free radical PE-AA^•^ that can further incorporate the oxygen molecule producing the peroxyl radical (PE-AA-OO^•^). This molecule is reduced to an anion and then protonated to PUFA-hydroperoxyl molecules (PE-AA-OOH/HpETE) in a step coupled to the regeneration of Fe^3+^. Non-heme iron is essential for the catalytic steps and redox cycling of LOX enzymes. These primary HpETE molecules can rapidly be converted by LOXs or other enzymes like GSH peroxidases or cyclooxygenases (COX1 and COX2) and cytochrome P450 (CYP450) family members, acting in series with LOXs [142], into end-products comprising (i) metabolites of arachidonic acid [12- or 15-hydroxyeicosatetraenoic acid (12-, 15-HETE)]; (ii) metabolites of linoleic acid [13- or 9-hydroxyoctadecadienoic acid (13-, 9-HODE)] and products derived from docosahexaenoic acid (C22:6, n-3), as well as (iii) the mentioned leukotrienes [141]. By producing PUFA-hydroperoxides in phosphatidylethanolamines, LOXs are considered pro-oxidative enzymes, and among the six functional isoforms existing in humans, 15-LOX1 and 15-LOX2 seem specifically involved in drug-induced ferroptosis [143]. The principal pro-ferroptotic molecules derived from arachidonic acid are represented by the downstream metabolites 15-hydroperoxyphosphatidylethanolamines (15-HpETE-PEs), considered also predictive biomarkers of ferroptosis [144,145] (Figure 2 and Figure 4).

The first implication of lipoxygenases in ferroptosis came from the group of Brent R. Stockwell [146]. Using a pool of siRNAs targeting all arachidonate lipoxygenases (A)LOX isoforms, the authors demonstrated that (A)LOXs are causally involved in the production of lethal lipid peroxides during ferroptosis triggered by erastin, a Xc- inhibitor, (but not by GPX4 inhibitors), albeit the specific isoforms can work in a context-dependent manner [146]. Furthermore, lipidomic analysis in ferroptosis-sensitive HT-1080 fibrosarcoma cells, under the same erastin conditions, indicated that loss of the polyunsaturated fatty acids was the most prominent change, accompanied by increased levels of ceramides and lysophosphatidylcholine (lyso-PC). Screening with representative PUFAs and MUFAs, cholesterol and cardiolipin, identified PUFAs as targets of lipid peroxidation [146]. Then, by screening phospholipids and their oxidation products, they found that double- and triple-oxygenated arachidonic acid and adrenic acid (AdA, C22:4, n-6), specifically incorporated into phosphatidylethanolamine (PE), can behave as signals for ferroptotic death [146]. Another study by Kagan et al. also showed that under RSL3-mediated GPX4 inhibition, AA and AdA are oxygenated mostly when incorporated into PE but not if present as free PUFAs [144]. Accordingly, Shah et al. demonstrated that supplementation with non-oxidizable fatty acids such as MUFAs or D-PUFAs (PUFAs containing deuterium atoms at the reactive bis-allylic sites) lowered ferroptosis [35,147]. As demonstrated by Shah et al., LOXs, albeit not indispensable during ferroptosis execution, are causal factors in the initiation of this process by contributing to the generation of a lipid hydroperoxide pool [35]. In that study indeed, overexpression of 5-LOX or 12-LOX, or 15-LOX1 isoforms was found to sensitize HEK293 cells to ferroptosis, and that only radical-trapping antioxidant compounds, like ferrostatin or liproxstatin-1, which protect membrane lipids from autoxidation, can prevent this type of cell death. Accordingly, it has been found that inhibition of several LOX isoforms either genetically or pharmacologically could prevent ferroptosis, at least to some extent [148].

As reported before, the incorporation of PUFAs into phospholipids, the principal substrates for LOX-mediated oxidation, is performed by two metabolic enzymes, ACSL4 and LPCAT3 (as discussed later in detail), and the ACSL3/LPCAT3/LOX constitute an important pro-ferroptotic cascade. The substrate competence of LOXs can be changed by their interaction with the tumor suppressor PEBP1 (phosphatidylethanolamine-binding protein, also called RKIP1), a scaffold protein inhibitor of protein kinase signaling, that consequently modulates the specificity of their products [149]. Wenzel et al. in their recent research further explored the role of this interaction in a ferroptosis context. By using Far Western blotting, cross-linking, and computational modeling, they demonstrated that PEBP1 specifically interacts/co-localizes with both 15-LOX isoforms and that the 15-LOX/PEBP1 complexes selectively generate the pro-ferroptosis molecule 15-HpETE-PE [150]. In search of selective inhibitors of the complex 15-LOX/PEBP1, it was very recently demonstrated that FerroLOXIN-1 and 2, screened from a customized library of 26 compounds, in vivo and in vitro could prevent ferroptosis provoked by RSL3 [151]. Of note, it has been reported that the 12-LOX enzyme is required for p53-mediated ferroptosis upon ROS stress [149]. However, lipid peroxidation by lipoxygenases is a controversial issue. Firstly, because the (A)LOX inhibitors used to confirm the (A)LOX role in ferroptosis are compounds with radical-trapping antioxidant activity [35], it remains unclear whether they prevent ferroptosis by inhibiting (A)LOXs or by trapping lipid peroxyl radicals. Secondly, data from the Cancer Cell Line Encyclopedia RNA-Seq indicate that (A)LOX mRNAs are present at very low levels and that (A)LOXs depletion in some cancer cells did not provide protection against ferroptosis [36] (Figure 4).

###### ACSL4 and LPCAT3 Enzymes

As mentioned before, membrane phospholipids, particularly phosphatidylethanolamines (PEs) containing PUFAs, are critical determinants in ferroptosis sensitivity [144]. ACSL4- and LPCAT3-mediated reactions are jointly required for the initiation of ferroptosis by controlling the availability of PUFA substrates for phospholipid synthesis (Figure 4). Specifically, ACSL4 catalyzes the addition of CoA to long-chain polyunsaturated-CoAs (PUFA-CoAs), preferentially of AA and AdA, therefore activating the biosynthesis of AA- or AdA-containing phospholipids (reviewed in [152]). PUFA-CoAs then become substrates of the lysophosphatidylcholine acyltransferase (LPCAT) enzymes, particularly LPCAT3, which esterifies the PUFA-CoAs into PUFA-phospholipid (reviewed in [153]). Pioneering studies demonstrated that ACSL4 is directly implicated in ferroptotic responses triggered by GPX4 inhibitors or by erastin [144,154,155]. In line with these results, it was reported that ACSL4 inactivation could block ferroptosis induced by FINs [36,156]. Doll et al. demonstrated that in ACSL4-knockout mouse embryonic fibroblasts, the amount of PUFA-phosphatidylethanolamines (PUFA-PEs) was reduced, and this protects cells against ferroptosis, whereas loss of other ACSL members failed in prevention [154]. This prompted the authors to further examine a panel of fatty acids to uncover specific substrates of ACSL4 and they found that AA- and AdA-containing PEs were selectively lowered in ACSL4-deleted cells, with unchanged levels of phosphatidylcholine and phosphatidylserine species [154]. Therefore, in the condition of GPX4 absence/inactivation, ACSL4 but no other ACSL family members could initiate the ferroptotic cascade by specifically activating AA- and AdA-PUFAs that, in turn, can become substrates for peroxidation mediated by LOX enzymes [144]. Indeed, ACSL4 is often used as a biomarker for ferroptosis induction. Since the main LPCAT3 substrates are PUFA-CoAs, the ACSL4/LPCAT3 axis increases the raw material for peroxidation reactions. Accordingly, inhibition of LPCAT3 can protect cells from ferroptosis by remodeling the PUFA content in PEs [157]. In agreement with this, Cui et al. recently reported that LPCAT3 and ACSL4 knockout can protect lung adenocarcinoma cells from ferroptosis, while ectopic expression displayed the opposite effect. In the same paper, it was demonstrated that LPCAT3 is transcriptionally regulated by the YAP/ZEB/EP300 pathway and that ACSL4 and YAP cooperate with LPCAT3 to establish ferroptosis sensitivity [158].

Hence, during ferroptosis, the activation of the ACSL4-LPCAT3-(A)LOX axis can mediate the production of lethal phospholipid hydroperoxides (15-HpETE-PEs) by reprogramming the PE profile. On the contrary, the ACSL3 isoenzyme that mostly produces MUFA-CoAs [159] and two ER membrane-bound acyltransferase isoenzymes, named MBOAT1/2 (membrane bound O-acyltransferase domain containing 1/2) [160] protect cells from ferroptosis by activating and producing MUFA-phospholipids, which are less oxidable substrates.

###### POR and CYB5R1 Enzymes

Searching for additional enzymes driving lipid peroxidation, two independent studies discovered that the oxidoreductase NADPH-cytochrome P450 reductase (POR) can trigger ferroptosis in most cancer cells [36,156]. This enzyme resides in endoplasmic reticulum (ER) compartments and donates electrons (from NADPH) to the cytochrome P450 (CYP) system or to other heme-proteins like heme oxygenase, cytochrome b_5_, and squalene monooxygenase using FAD/FMN as a cofactor [161] (Figure 4). Both studies used CRISPR screening to identify genes whose inactivation suppressed FIN-triggered ferroptosis, highlighting as top suppressor hits the cytochrome P450 reductase and the pro-ferroptotic ACSL4 (already identified in other studies). Accordingly, a deficiency of POR promoted resistance to ferroptosis triggered by different ferroptosis-inducing agents, named FINs, in a wide range of cancer cells and lowered lipid peroxide levels without affecting the amounts of GPX4 or glutathione and cell phospholipid profiles [36,156]. Yan et al. demonstrated that the electron transfer activity of POR is essential for ferroptosis albeit lipid peroxidation occurs independently of CYPs, and Zou et al. suggested that POR can push Fenton reactions during electron transfer to CYPs with consequent lipid peroxidation [36,156]. Furthermore, Yan et al. identified another oxidoreductase, namely the NADH-cytochrome b5 reductase 1 (CYB5R1), whose deficiency inhibited lipid peroxidation/ferroptosis. CYB5R1 deletion showed modest effects compared to POR inactivation, but the combination of both POR and CYB5R1 deficiency provoked ferroptosis resistance [156]. Since POR and CYB5R1 are involved in redox metabolism, Yan et al. supposed that these enzymes could generate ROS, likely during electron transfer reactions. In fact, the authors proved that purified enzymes, especially POR, produce H_2_O_2_ in an NADPH- and oxygen-dependent manner and that POR-deleted cells have reduced levels of H_2_O_2_ [156]. These experiments demonstrate that POR and CYB5R1 (albeit more moderately) can promote lipid peroxidation and ferroptosis through the production of H_2_O_2_.

###### FAR1 Enzymes

Fatty acyl-CoA reductase 1 (FAR1) is a peroxisomal enzyme essential for supplying fatty alcohols during the biosynthesis of ether phospholipids (ePLs) (Figure 4). There are two classes of ether PLs: alkyl-ether PLs and vinyl-ether PLs (also known as plasmalogens) with important cellular roles, including antioxidant defenses, signaling, and structural functions [162]. Ether PL synthesis initiates in peroxisomes, terminates in the endoplasmic reticulum (ER), and requires fatty alcohols provided by the FAR enzymes. Peroxisomes, now considered crucial contributors to ferroptosis [163], are membrane-bound organelles with multiple functions, including H_2_O_2_ production and elimination, synthesis of certain lipids, and degradation of long- and branched-chain fatty acids. Recent evidence showed that the plasmalogen subclass of ether-linked phospholipids could be involved in ferroptosis (FAR1-alkyl-ether lipids axis), thus providing an additional source of PUFA-containing phospholipids, independently of those involved in the ACSL4/LPCAT3 axis (reviewed in [163,164]). In a first work, using a CRISPR-Cas9 suppressor screen, Zou et al. uncovered novel genes comprising alkylglycerone phosphate synthase (AGPS), fatty acyl-CoA reductase 1 (FAR1), glyceronephosphate O-acyltransferase (GNPAT), and 1-acylglycerol-3-phosphate O-acyltransferase 3 (AGPAT3) involved in the synthesis of plasmalogens, as top pro-ferroptosis genes [165]. By lipidomic profiling, the authors demonstrated that the synthesis of polyunsaturated ether PLs can ultimately lead to lipid peroxidation during the pro-ferroptotic cascade [165]. Next, by metabolite screening, Cui et al. identified 1-hexadecanol (1-HE) as a molecule able to trigger ferroptosis under GPX4 inhibitor treatments [166]. 1-HE originates from the reduction of palmitic acid (C16:0) to fatty alcohol (the rate-limiting process in plasmalogen synthesis) through the peroxisome-localized enzyme called FAR1. Cells treated with 1-HE exhibited a high level of lipid peroxidation and FAR1 deletion increased resistance to erastin and RSL3-induced ferroptosis of HT1080 cells [166]. Of note, FAR1 gene transcription is induced upon ER stress [167] through ATF6, a sensor/effector of the unfolded protein response (UPR) that can also upregulate the peroxisomal antioxidant enzyme catalase.

### 3.2. Antioxidant Defenses and Lipid Peroxidation

The preceding part highlights the critical role of lipid peroxidation (either induced by ROS/RNS or by enzymatic activities) in the ferroptosis process. However, sophisticated endogenous antioxidant systems in living cells can counteract the formation of ROS/RNS and/or lipid hydroperoxides. Antioxidant defenses preserving redox homeostasis may be grouped into different categories able to: (i) prevent the formation of active oxidant molecules by scavenging or quenching ROS/RNS producing reactions; (ii) remove the active (lipo)oxidant products and repair (lipo) damages; and (iii) induce adaptive responses through activation or inhibition of specific enzymes/pathways [94,168]. Dysfunctions of the antioxidant systems are linked to ferroptosis. As noted above, along with iron accumulation and/or metabolic perturbations, excessive RNS/ROS production due to the failure of redox homeostasis can foster biochemical pathways leading to the unrestrained generation of lipid radical species. By selectively reducing hydroperoxy-PEs, the selenoperoxidase GPX4 represents a crucial enzymatic system to prevent lethal lipid hydroperoxides; however, other novel antioxidant systems that operate in different compartments have been unveiled. In the next sections we will focus on those systems implicated in defenses against ferroptosis.

#### 3.2.1. Mechanisms That Prevent or Intercept ROS/RNS and Lipid Peroxides

Considering general defenses against ROS/RNS increases, the isoenzymes SOD1-3 are directly implicated in the dismutation of O_2_^•−^ into H_2_O_2_, hence limiting potential damages in different cell compartments, including membranes [169]. However, the role of these enzymes has been under-investigated in ferroptosis. Mitochondrial superoxide dismutase (SOD2) and manganese-dependent superoxide dismutase (MnSOD) work in the mitochondrial matrix and a recent paper demonstrated that SOD2 can protect mitochondrial membrane lipids from ROS-mediated oxidation in nasopharyngeal carcinoma cells [170]. SOD2 knocked-down cells exhibited higher concentrations of O_2_^•−^ that consequently push membrane lipid peroxidation/ferroptosis and increased their radiosensitivity. Because the main effects of SOD2 deficiency are on the ETC complexes I and II while the enzyme dihydroorotate dehydrogenase (DHODH) contributes to complex III activity [171,172,173], the absence of both SOD2 and DHODH can reduce ferroptosis and radiotoxicity, possibly due to a deficiency of O_2_^•−^ production.

Albeit both O_2_^•−^ and H_2_O_2_ possess a relatively low reactivity, H_2_O_2_ represents a recognized source of HO^•^ and HOO^•^ that can initiate lipid peroxidation events. H_2_O_2_ can be decomposed by the peroxisomal enzyme catalase (the most used) and glutathione peroxidases (GPXs), as well as by peroxiredoxins (PRXs) (reviewed in [174]). Catalase can have a protective function against ferroptosis; indeed, its deficiency has already been linked to many oxidative stress-driven diseases (reviewed in [175]). Regarding ferroptosis, it has been demonstrated by Hwang et al. that up-regulation of catalase mediated by PPARδ, a transcription factor implicated in lipid metabolism and energy homeostasis, is responsible for ferroptosis resistance in mouse embryonic fibroblasts (MEFs) derived from cysteine/glutamate transporter (xCT)-knockout mice [176].

NO^•^ is reported to be an important antioxidant defense, albeit frequently considered a pro-oxidant molecule. Being present not only in low-density lipoproteins (LDLs) in vascular walls but also in cell membranes, it can directly react with lipid-derived peroxyradicals (LOO·) and therefore terminate radical chain propagation reactions [177]. The anti-ferroptosis properties of NO^•^ have been recently reported by Kapralov et al. [178]. They demonstrated that RSL3-induced ferroptosis in RAW 264.7 (bone marrow-derived macrophages) and microglial cells can be inhibited by elevated levels of inducible NO synthase (iNOS/NOS2) that consequently boosts NO^•^ production. Furthermore, manipulation of iNOS levels or iNOS pharmacological inhibition, as well as treatments with NO^•^ donors, supported the protective role of NO^•^, which counteracted ferroptosis by nitroxygenation of intermediates of the ACSL4-LPCAT3-LOX15 axis, and this effect was specifically dependent on the complex LOX15/PEBP1 (yielding the lethal 15-HpETE-PEs) [145]. Another paper by Homma et al. reported that under ferroptotic conditions triggered by cysteine depletion, GPX4 inhibition, or oxidative treatments in mouse hepatoma Hepa 1–6 cells, the NO^•^ donor NOC18 suppressed ferroptosis, which is probably mediated by inhibition of ROS production and termination of the lipid peroxidation cascade [179].

Removal of membrane-embedded oxidized lipid species, the essence of ferroptosis, is principally mediated by the Ca^2+^-independent phospholipase A2β (iPLA2β) belonging to the PLA2 family (reviewed in [180]). This enzyme has been implicated in protective processes because of its catalytic activity toward 15-HpETE-PEs: indeed, naturally occurring mutations in its gene (PNPLA9) are associated with neurodegenerative diseases like neurodegeneration with brain iron accumulation (NBIA) class [181] and Parkinson disease 14, autosomal recessive (PARK14) [182]. Previous studies supported the role of this enzyme in the hydrolysis of peroxidized phospholipids (reviewed in [183]), which consequently can counteract ferroptosis cell death, independently of GPX4, as proved in more recent studies [184,185]. Interestingly, Sun et a. found that fibroblasts derived from a patient with a mutation in the PNPLA9 gene and iPLA_2_β-deficient cells exhibited higher levels of toxic 15-HpETE-PE compared to wild-type cells upon RSL3 treatment [186]. In that study the authors demonstrated that the activity of iPLA_2_β enzyme is linked to LPCAT3-mediated remodeling of membrane PLs, producing AA-PE required for pro-ferroptotic molecule generation. In addition, a parkinsonian phenotype was documented in CRISPR-Cas9-engineered mice carrying a mutant PNPLA9 gene that displayed a 15-HpETE-PE increase, which also predominated in the midbrains of rotenone-infused parkinsonian rats [186]. Another study identified iPLA2β as an antagonist of the p53-driven ferroptosis cascade upon exposure to tert-butyl hydroperoxide (TBH), a ROS generator [184]. It was demonstrated that ferroptosis triggered by p53 was independent of the GPX4–ACSL4 axis and that iPLA2β was indeed a p53-dependent gene. Depletion of endogenous iPLA2β increased ROS-mediated ferroptosis in wild-type p53 cells, but not in p53 mutated cells. In vivo experiments in human melanoma A375 xenograft tumors further supported that loss of iPLA2β can push p53-mediated ferroptosis. Molecular analyses showed that iPLA2β downregulates ALOX12-generated peroxidized membrane lipids [184].

#### 3.2.2. Enzymatic Mechanisms That Protect against Lipid Peroxidation

Protection against lipid peroxidation is mainly represented by the redox-related enzymatic system, which could be implicated in the repair of damaged lipids and/or in the elimination of (lipo)oxidation products. This involves the antioxidant pathways of GPX4/glutathione/cysteine, peroxiredoxin 6 (PRDX6), the mevalonate pathway, FSP1/coenzyme Q_10_ (CoQ_10_), microsomal glutathione transferase 1 (MGST1), mitochondrial GPD2/DHODH/CoQ_10_, and the GCH1/tetrahydrobiopterin (BH_4_) system that can also act in concert with other defenses able to fine-tune the lipid composition (Figure 5).

##### GPX4/Glutathione/Cysteine

Glutathione peroxidases (GPXs) belong to the selenoprotein family, containing selenocysteine (Sec) in their redox active motifs [187]. These enzymes reduce hydroperoxides, including H_2_O_2_, to their alcohol derivates using two glutathione molecules that are consequently oxidated to glutathione disulfide (GSSG) [188]. In humans, there are eight GPX isoforms, with five enzymes (GPX1-4, GPX6) that are selenoproteins harboring Sec residues and one, GPX5, with cysteine (reviewed in [189]). They display distinct localizations: all isoforms are cytosolic; some localize also in the nucleus (GPX1-2 and GPX4) or in the mitochondria (GPX1 and GPX4). Furthermore, GPX3 is secreted from cells and GPX4 is present within the cell membranes. GPX4, originally called phospholipid hydroperoxide GPX (PHGPX4), directly reduces peroxidized phospholipids (PLOOH), even those present in membranes, to lipid alcohol (PLOH) and it is now considered an integral membrane system of resistance to oxidative stress-induced PLOOH [190]; of note, this activity could not be replaced by other GPXs or redox-active enzymes [191]. GPX4 harbors eight nucleophilic amino acids (one selenocysteine and seven cysteines); thus, it can react with numerous electrophiles [146]. Schnurr et al. provided the first evidence that GPX4 activity in a reconstituted model system can be coupled to reducing hydroperoxy ester lipids generated by LOX15, one isoform of the lipoxygenase family [192], which, as reported above, can produce pro-ferroptotic 15-HpETE-PE molecules. In 2003, other papers reported that GPX4 is an essential antioxidant enzyme [193,194], demonstrating that GPX4 knockout mice were embryonic lethal and that cells lacking GPX4 produced high levels of lipid hydroperoxides and were highly sensitive to oxidative stress [195,196]. Another study demonstrated that α-tocopherol (α-Toc), the major antioxidant in membrane compartments, can rescue cell lethality due to GPX4 deficiency [197]. In that paper, it was also demonstrated that cell death in GPX4-knockout cells was significantly accelerated by arachidonic acid and linoleic acid and that functional 12/15-LOX enzymes were required for death induction since a general lipoxygenase inhibitor prevented cell death. Further analyses implicate the apoptosis-inducing factor (AIF) as a mediator of this type of cell death [197]. Thus, inactivation of the GPX4 system seems to be generally implicated in different types of cell death, like apoptosis, necroptosis, and pyroptosis, as well as ferroptosis, suggesting a possible context-related role of GPX4 [189]. Regarding ferroptosis, in 2014, a paper was published demonstrating that GPX4, when overexpressed, represents a crucial anti-peroxidant system, able to reduce the process of ferroptosis induced by different compounds called ferroptosis inducers (FINs) [198]. At the genetic level, the principal direct evidence that GPX4 knockdown pushes ferroptotic-related cell death comes from the work of Angeli et al. [199]. In this paper, the authors used inducible GPX4 (−/−) mice and demonstrated the essential role of the GPX4/glutathione axis in preventing lipid peroxidation; indeed, mice lacking GPX4 undergo acute renal failure and early death: a clear initial implication of ferroptosis in pathological processes [199]. Moreover, in the cancer field, direct inhibitors of GPX4 expression/activity have been actively searched and many compounds, now called Class-II FINs, have been proposed thus far (e.g., ML162, ML210, FIN56, FINO2, and RSL3), with promising results in vitro and in vivo [198,200]. The rationale behind this is based on the fact that cancer cells normally exhibit a basal increase in ROS/RNS levels and this condition is essentially dependent on multiple scavenging systems, and thus, a further increase in ROS/RNS production renders cancer cells more sensitive to anticancer drugs driving oxidative stress [201,202].

Both GSH and cysteine, the amino acid precursor of GSH, can coordinate antioxidant GPX4 activity during ferroptosis. Pioneering studies by Dixon et al. firstly linked GSH/thiol metabolism to ferroptosis [11]. Glutathione is an obligate cofactor for GPX4 and mounting evidence suggests that GSH biosynthesis as well as cysteine imports and/or synthesis critically mediates the response of GPX4 to ferroptosis inducers (reviewed in [133]). The catalytic cycle of GPX4 indeed implicates a first GSH-dependent reduction of the reactive selenic acid (-SeOH) generated by a reaction between the peroxide moiety and the GPX4-selenol (-SeH) group, creating the intermediate GPX4-selenide disulfide (-Se-SG). A second GSH is then used to regenerate the GPX4-SeH group and the oxidated GSH is released as GSSG [191]; therefore, two glutathione are necessary in the catalytic cycle. Consistent with this, processes and/or drugs that specifically enhance intracellular GSH levels can rescue ferroptosis cell death, whereas depletion of GSH is an important contributor to death initiation and execution [133].

Here, we briefly discuss GSH and cysteine metabolism as was also discussed in other recent reviews [133,203,204]. The tripeptide GSH containing Glu, Cys, and Gly amino acids is produced in a two-step process involving two different enzymes, glutamate-cysteine ligase (GCL), which is composed of a catalytic γ-glutamylcysteine synthetase (γ-GCS) and modifier (GCLM) subunits, and glutathione synthase (GSS). The rate-limiting enzyme of GSH synthesis is γ-GCS, involved in the first step, and it has been found that in some conditions, buthionine sulfoximine (BSO) can induce ferroptosis by inhibiting this enzyme [198]. The GSH antioxidant activity is due to its essential role as a cofactor for multiple antioxidant proteins, including GPXs, GSH S-transferases, and glutaredoxins [205]. Furthermore, GSH depletion can trigger labile iron increase: this can be related to the fact that GSH can bind free iron and this complex is then recruited by the adaptor protein poly(rC) binding protein 1 (PCBP1) that delivers ferrous iron to ferritin for storage [206,207].

Cysteine (Cys) availability regulates the de novo synthesis of GSH [101,208]. One of the major systems that increase the intracellular amounts of cysteine is represented by the Xc- system, a Na^+^-independent cystine/glutamate antiporter. This is a transmembrane heterodimer, composed of two subunits (SLC7A11 and SLC3A2) linked by disulfide, that can exchange a molecule of cystine, the dipeptide precursor of cysteine, with one of glutamate, in an ATP-dependent manner [11,209,210]. Imported cystine is then reduced to cysteine by the action of specific cytosolic cystine reductases, including thioredoxin 1 (TRX1) and a specific thioredoxin-related protein (TRP14) [211]. A downstream consequence of system Xc- inhibition is a rapid drop of GSH levels causing ferroptosis/cell death in most cancer cells [11,48,107,212] and tumor suppression [213]. Indeed, Xc- system represents a predominant route for the cystine import process, and numerous papers demonstrated that increased SLC7A11 expression/function enables cells to survive under different types of stressful conditions, preserving the GSH content and redox homeostasis (reviewed in [214]). Accordingly, most of the Class-I FINs, such as erastin and many analogs like sulfasalazine and sorafenib that block the Xc- system via direct SLC7A11 targeting, trigger ferroptosis, as a consequence of disabled GSH/GPX4 function [48,215]. However, more recently, Yan et al. demonstrated that different expression levels of SLC7A11 dictate protection or death under H_2_O_2_ treatments in glucose-depleted cancer cells, with low levels associated with protection and high levels with death. In this latter condition, toxicity may rely on NADPH depletion (required for cystine reduction) and disulfide stress, potentially due to cystine and GSSG increase. In vivo data further demonstrated that high SLC7A11 promoted primary tumor growth but repressed tumor metastasis [216]. In addition, cells can import cystines derived from the γ-glutamyl cycle in which cysteine (readily oxidated to cystine) is liberated from extracellular glutathione through the activities of the two enzymes γ-glutamyl transpeptidase (GGT) and dipeptidase (DP) [217]. Very recently, GGT1 has been implicated in modulating ferroptosis in glioblastoma [218]. In fact, cystine deprivation-induced ferroptosis was favored by GGT1 inactivation (pharmacological inhibition or deletion) in high-density cells and viability was restored through a supply of cysteinyl-glycine, the GGT1 cleavage product [218]. In another work, over-expression of GGT1 restrained ferroptosis and autophagy induced in retinal ganglion RGC-5 cells by oxygen-glucose deprivation/reoxygenation treatment and this was mediated by a direct interaction with the GCLC enzyme [219]. It is important to note that aside from GSH, cysteine is used for the biosynthesis of proteins, iron–sulfur clusters, and heme, as well as the production of taurine and coenzyme A metabolites, raising the possibility that the observed antiproliferative effects due to cystine depletion could also be related to a decrease of such crucial molecules [195,203,204,220,221]. Cysteine could be also exported out of the cells through neutral amino acid transport systems, such as SLC38A2 [222], which indicates that a low cysteine level in these cells is dynamically controlled [212].

The transsulfuration pathway (TSP) represents an important metabolic cascade implicated in the de novo biosynthesis of cysteine [223]. This pathway produces cysteine from homocysteine, a molecule originating from dietary methionine, an essential amino acid. In mammals, this metabolism represents a unique endogenous source of cysteine, and the reactions involving the transfer of sulfur to serine are catalyzed by two enzymes, cystathionine β-synthase (CBS) and cystathionine γ-lyase (CGL/CSE) [224]. By condensing homocysteine with serine, CBS generates the intermediate cystathionine, which becomes the substrate for CGL and liberates cysteine, α-ketobutyrate, and NH_4_^+^. The importance of the TSP pathway was discovered by the Stockwell group through siRNA screening for genes involved in suppressing erastin-induced ferroptosis. They identified the cysteinyl-tRNA synthetase (CARS) as a gene knockdown that can prevent erastin-induced ferroptosis in different cell contexts [225]. The absence of CARS inhibited the generation of lipid ROS in erastin-treated cells, without affecting ferroptosis triggered by GPX4 inhibitors (RSL3 and FIN56) or influencing iron homeostasis. Indeed, CARS knockdown prevents ferroptosis through activation of the TSP pathway by increasing intracellular cysteine and supplementing the glutathione content (GSH and GSSG); this consequently supports GPX4 activity, thus rescuing in turn erastin-induced ferroptosis [225]. Also, Wang et al. demonstrated that inactivation of CBS by a specific inhibitor (named CH004) induced ferroptosis in HepG2 cells and in liver tumor xenograft mouse models [226]. Other recent papers have linked the transsulfuration pathway to ferroptosis resistance. For example, the protein DJ-1/PARK7, an anti-oxidative stress gene, protects cancer cells from ferroptosis by preserving the activity of S-adenosyl homocysteine hydrolase, a hydrolase specific for S-adenosyl-L-homocysteine that produces homocysteine, hence maintaining cysteine synthesized from the TSP pathway by fueling GSH synthesis [227]. DJ-1 has also been implicated in mediating the NRF2/GPX4 signaling pathway to prevent ferroptosis [228]. Of note, cystathionine is also a substrate of system Xc- activity [229] and can protect against oxidative stress, especially in the immune system [230]. In general, the TSP pathways and downstream cysteine have multifaced roles, as recently reviewed elsewhere [223]. In conclusion, the GPX4/glutathione/cysteine axis represents a major determinant for defenses against ferroptosis and transcription factors, like NRF2, ATF4, and JUN (as discussed in the next section), that have emerged as important regulators of antioxidant-defenses in ferroptosis by controlling the systems that replenish GSH content.

##### Peroxiredoxin 6 (H_2_O_2_ and Lipid Peroxides Decomposition and Repair of Membrane)

Another negative regulator of ferroptosis is peroxiredoxin 6 (PRDX6), belonging to the family of non-selenium peroxidases (PRDx), which hampers lipid peroxidation by using three enzymatic activities, peroxidase, phospholipase A2, and acyl transferase [231]. The PRDX family comprises six members grouped into two classes based on the number of conserved cysteine residues in their catalytic site: PRDX1-5 (two Cys) and PRDX6 (one Cys) [232]. PRDXs remove the build-up of H_2_O_2_ or other organic hydroperoxide. To reduce substrates, PRDX6 uses GSH as a physiological reductant, while PRDX1-5 enzymes form disulfide bonds, which are subsequently reduced via the thioredoxins (TRXs) systems [174]. As mentioned above, PRDX6 can mediate lipid repair through three functions: (i) direct reduction of PLOOH to alcohol (PLOH) using the GSH-dependent PRDX6/PHGP activity, like GPX4; (ii) removal of the LOOH moiety using a unique Ca^2+^-independent phospholipase A_2_ activity (PRDX6/PLA_2_) toward phosphatidylcholine peroxides (PCOOH), producing lysoPC; and (iii) reacylation of lysoPC with reduced FA-CoA by an intrinsic coupled LPCAT activity (PRDX6/LPCAT) [231,233,234,235]. Lu et al., using inducible knockdown of PRDX6 in H1299 cells, recently demonstrated that PRDX6 has a protective role against ferroptotic stresses induced by erastin and RSL3 [236]. This effect was documented to be GSH-independent since PRDX6 utilizes its PLA2 activity to hydrolyze the LOOH moiety. Furthermore, PRDX6 knockdown stimulates the expression of heme oxygenase 1 (HO-1) through NRF2 transcriptional control and this enhanced HO-1 activity can consequently push ferroptosis by releasing iron [236]. In a very recent work, Liao et al. [237] demonstrated that PRDX6 ectopic expression significantly inhibited pulmonary hypertension (PH), a cardiopulmonary disorder, both in vitro in pulmonary endothelial cells (PAECs) and in vivo in a rat PH model. The authors discovered that PRDX6 was expressed at lower levels in PH models, and this was associated with decreased expression of GPX4 and FTH1 as well as with increased NOX4. When PRDX6 was overexpressed, the GPX4 and FTH1 levels were restored while NOX4 decreased. The authors also tested the levels of HMGB1 (high mobility group box 1) in the PAEC supernatant, since previous studies indicated that this protein is released by ferroptotic cells activating the inflammatory pathway (reviewed in [238]). In fact, it was demonstrated that PRDX6 reduced HMGB1 levels that were increased under RSL3 treatment, whereas analysis of the TLR4/NLRP3 inflammasome signaling pathway downstream of HMGB1 confirmed that PRDX6 can control the release of HMGB1 and consequently the activation of TLR4 in macrophages [237]. Actually, the protective role of PRDX6 against ferroptosis has been proven to be important in diabetic kidney diseases [239], inflammatory bowel disease [240], and periodontitis [240]. Finally, in another recent study, the protein levels of PRDX6 (and ACSL3) were found to be higher in tumor tissues of lung adenocarcinoma, implicating a possible role of PRDX6 in ferroptosis resistance in lung cancer cells [241]. Moreover, decreased PRDX6 (and GPX4) expression after knockdown of NOTCH3, a signaling membrane receptor, could initiate ferroptosis by increasing ROS/lipid peroxidation and iron in non-small cell lung cancer cells [242].

##### HMGCR/Mevalonate Pathway (Production of IPP and CoQ_10_)

The mevalonate biosynthetic pathway (also known as the isopredoid pathway) starts from the condensation of three acetyl-CoAs, forming mevalonate, by the enzyme hydroxymethylglutaryl-CoA reductase (HMGCR). This pathway is linked to ferroptosis because it represents the primary source of the isopentenyl pyrophosphate (IPP), an essential intermediate molecule for at least two pathways: biosynthesis of selenoproteins, including GPX4 [191], and the synthesis of metabolites comprising coenzyme Q_10_, (CoQ_10_), sterols, and vitamin K (VK) [243]. The rate-limiting step of the mevalonate pathway is catalyzed by HMGCR and inhibiting HMGCR could induce ferroptosis, at least in some cancer cells [244]. Concerning selenoproteins, IPP is the precursor of its isomer dimethylallyl diphosphate, a substrate for the tRNA isopentenyltransferase1, a key enzyme participating in the production of Sec-tRNA, a complex process necessary to insert the selenocysteine in the catalytic center of selenoproteins [245,246].

In 2016, by evaluating about 3000 compounds Shimada et al. discovered a novel selective ferroptosis inducer, designated as FIN56 [247]. In search of a molecular mechanism, the authors investigated possible death pathways mediated by FIN56, reporting that FIN56 did not affect GSH levels but rather it lowered GPX4 protein levels, independently of transcriptional or translational events. FIN56-induced ferroptosis relies on the direct binding and activation of squalene synthase (SQS) protein, acting downstream of the mevalonate pathway to produce various metabolites. Testing of metabolite species revealed that only idebenone, a hydrophilic analog of CoQ_10_, was able to rescue FIN56-induced ferroptosis, thus indicating a negative effect of FIN56 on the mevalonate-derived ubiquinone/coenzyme Q_10_ [247]. Coenzyme Q_10_ is a redox-active benzoquinone bearing a polyisoprenoid tail, which renders this molecule extremely hydrophobic. It serves as an electron carrier in the ETC, and as cofactor of various oxidoreductases and dehydrogenases in different cellular processes like pyrimidine synthesis, fatty acid oxidation, and sulfide oxidation. In the ferroptosis context, non-mitochondrial CoQ can behave as a lipophilic antioxidant capturing free radicals and supporting enzymatic reactions in cell membranes [248]. The protective role of CoQ_10_ in ferroptosis will be discussed in the next section.

##### FSP1/Coenzyme Q_10_ (GSH-Independent)

The flavoprotein NAD(P)H-dependent oxidoreductase (FSP1) belongs to the family of type II nicotinamide adenine dinucleotide-H (NADH): quinone oxidoreductase that reduces quinones to the corresponding hydroquinone, using NAD(P)H. A protective system independent of glutathione, comprising FSP1 and using CoQ_10_ as a free radical scavenger for substrate/s, was simultaneously identified by two groups [249,250]. The authors independently reported the identification of a gene named “ferroptosis suppressor protein 1” (FSP1), previously known as “apoptosis inducing factor mitochondria associated 2” (AIFM2), which was able to confer protection against ferroptosis induced by GPX4 inactivation (by RSL3) [250] or by genetic deletion [249]. The authors proposed that FSP1 can scavenge phospholipid peroxyl radicals by increasing the steady state of reduced coenzyme Q_10_ (CoQH_2_)/ubiquinol, which acts as a lipophilic antioxidant. This protein associates with plasma and endoplasmic reticulum (ER) membranes, as well as to the outer side of the mitochondrial inner membrane, and represents a potent GSH-independent ferroptosis suppressor system, as discussed in recent reviews [251,252]. It has been demonstrated that FSP1 activity is also important for other anti-ferroptotic mechanisms, namely through the nonclassical redox cycle of vitamin K [253], with FSP1 acting as NAD(P)H-dependent vitamin K reductase, producing vitamin K hydroquinone (VKH2), a scavenger that also prevents lipid peroxidation [253].

With the aim to better understand the structural/catalytic properties of FSP1 in ferroptosis resistance, a recent paper demonstrated in vitro that FSP1 adopts a glutathione reductase-like fold, functions as a dimer with a myristoyl anchor, and that the presence of 6-hydroxy-FAD is required for its CoQ oxidoreductase activity, which in turn can behave as a free radical scavenger [254]. In general, FSP1 potentially suppresses ferroptosis by generating CoQH_2_ on the cell membranes. Other investigations associated the FSP1-mediated protection with the activity of ESCRT-III (endosomal sorting complexes required for transport III) complex, independent of ubiquinol [255]. ESCRT-III is one of the five cell complexes involved in membrane remodeling and fission processes and it had previously been implicated in the repair of damaged plasma membranes triggered during ferroptosis [256] or other forms of cell death [257]. In particular, the authors showed that specific proteins present in the complex, CHMP5 and CHMP6 (charged multivesicular body protein), are important components in driving ferroptosis resistance. In another study, Yoshioka et al. found that FSP1 can prevent ferroptosis by fostering the activity of ESCRT-III [258]. Very recently, a study by Zhang et al. demonstrated that FSP1 specifically requires NADPH and not NADH for its protective function in ferroptosis [259]. This is different from the AIFM1 isoenzyme that specifically interacts with NADH. Since FSP1 is regulated by the NRF2-Keap1 pathway, recent evidence suggests that in cells resistant to ferroptosis due to Keap1 inactivation, like mutant lung cancers, the genetic deletion of NRF2 or FSP1 as well as the pharmacological inhibition of CoQ biosynthesis with 4-chlorobenzoic acid (4-CBA) could represent a novel therapeutic strategy to combat cancer growth [260,261].

##### Mitochondrial DHODH/GPD2/CoQ_10_ (GSH-Independent)

Novel mechanisms of cellular defense against ferroptosis involve two mitochondrial enzymes: the dihydroorotate dehydrogenase (DHODH) and the glycerol-3-phosphate (G3P) dehydrogenase 2 (GPD2) [119,120], both located in the inner mitochondrial membrane that operates in parallel [262,263]. The ETC enzyme DHODH catalyzes the oxidation of dihydroorotate to orotate by two sequential redox reactions using first FMN and then ubiquinone, acting as an ultimate electron acceptor (CoQH_2_). In their first work, Mao et al. unveiled through global metabolomic analyses under GPX4 inhibitors (RSL3, ML210, and ML162) the depletion of C-Asp, an intermediate of de novo pyrimidine biosynthesis, with a parallel increase of uridine, the final metabolite [119]. Accordingly, RSL3 treatment increased the DHODH activity and DHODH inhibition induced mitochondrial lipid peroxidation, especially in cells with low expression of GPX4. Notably, DHODH deletion did not influence GSH levels or the expression of GPX4, SLC7A11, and ACSL4, indicating an independent mechanism. The proposed model is that DHODH suppresses mitochondrial lipid peroxidation and ferroptosis, acting in parallel to mitochondrial GPX4, but not with cytosolic GPX4 or FSP1 [119]. Because DHODH could couple dihydroorotate oxidation to CoQ reduction, increased DHODH activity in turn implements CoQH_2_, thus preventing mitochondrial lipid peroxide [119]. Another study of the same research group found G3P as another metabolite that was reduced upon treatments with all three GPX4 inhibitors [120]. G3P is oxidized to dihydroxyacetone phosphate (DHAP) by the enzyme GPD2, which donates electrons to the ETC, using FAD as a cofactor, producing CoQH_2_. The authors proved that G3P supplementation can inhibit RSL3-induced ferroptosis in different cancer cell lines and that GPD2 deletion significantly sensitized cells to GPX4 inhibitors, and this was accompanied by increased mitochondrial lipid peroxidation. Since GPD2 deletion did not sensitize cells to FIN56, a compound that also depletes CoQ (besides GPX4), and considering the role of CoQ in restraining lipid peroxidation/ferroptosis, the authors investigated if GPD2 antagonizes ferroptosis in a CoQ-dependent manner. In fact, inhibition of CoQ biosynthesis either pharmacologically with 4-CBA or by deletion of the key biosynthetic enzyme COQ_2_, increased lipid peroxidation and ferroptosis. Furthermore, genetic interaction studies demonstrated that GPD2 in mitochondria works in parallel with DHODH and with GPX4 to protect against ferroptosis. Finally, GPD2 deletion can synergize with GPX4 deletion to suppress tumor growth in vivo [120]. In conclusion, both papers reported GPX4-independent antioxidant pathways involving ubiquinone that suppress mitochondrial lipid peroxides.

##### GCH1/Tetrahydrobiopterin (BH_4_)

The GTP cyclohydrolase 1 (GCH1) enzyme is an essential component in the de novo synthesis of tetrahydrobiopterin (BH_4_) by converting GTP into D-erythro-7,8-dihydroneopterin triphosphate, a precursor of BH_4_, an important cofactor for NOS and hydroxylase enzymes. The BH_4_ pathway has recently emerged as another potent GSH-independent antioxidant mechanism, important in ferroptosis resistance; BH_4_ indeed can behave as an endogenous radical trap in lipid membranes [208,264]. In their paper, by performing a genome-wide activation screen, Kraft et al., identified numerous genes from cells resistant to different ferroptosis inducers. Intersection of the data from all resistant cells unveiled the GTP cyclohydrolase 1 (GCH1) gene as a specific common ferroptosis antagonist, but unable to protect cells from other types of death. GCH1 catalyzes the rate-limiting step of pterins biosynthesis, such as BH_4_. Metabolomic analyses revealed that GCH1 overexpression increases BH_4_ and pteridines from the folate biosynthetic pathway, like dihydrobiopterin (BH_2_), an oxidized form of BH_4_. Furthermore, analysis of lipids extracted upon GCH1 overexpression indicates that cells are protected from the degradation of specific phosphatidylcholines with two PUFAs chains, associated with increased levels of CoQH_2_ (the BH_4_–CoQH_2_ axis). Thus, downstream metabolites of the GCH1 cascade act to suppress lipid peroxidation and ferroptotic cell death [264]. In another paper, Soula et al., by using a metabolism-focused library of sgRNAs targeting ~3000 genes, identified novel genes essential for survival in Jurkat and Karpas-299 cells upon either cystine depletion (by erastin) or GPX4 inhibition (by RSL3) [208]. Not surprisingly, these analyses revealed that common scoring protective genes encompass glutathione biosynthesis, whereas other genes were specific for each inhibitor. In particular, in the RSL3-dependent GPX4 inhibition, a predominant protective pathway engages the tetrahydrobiopterin biosynthesis that correlates with increased levels of GCH1. For erastin resistance, one top-scoring gene was represented by sideroflexin 1 (SFXN1), encoding the mitochondrial transporter of serine. A more detailed analysis in SFNX1 knockout cells also identified high levels of other mitochondrial metabolites, like acetyl-coenzyme A, taurine, hypotaurine, and two sulfur cysteine-derived metabolites. Furthermore, in these cellular systems, it seems that the anti-ferroptotic function of BH_4_, which synergizes with α-tocopherol, is related to its capacity to trap radicals in membrane lipids. In particular, BH_4_ antioxidant activity is linked to the activity of dihydrofolate reductase (DHFR), which can regenerate BH_4_ from BH_2_ (the BH_4_–DHFR axis) [208]. The recent work of Hu et al. searched for other potential links of the GCH1/BH_4_ cascade with ferroptosis and demonstrated that ablation of GCH1 increased the amounts of both cytosolic and mitochondrial Fe^2+^ upon erastin treatment [265]. This effect was linked to an enhanced NCOA4-mediated ferritinophagy, the specificity of which was tested by using 3-methyladenine (3MA), a compound that inhibits autophagy. 

Moreover, Wang et al. discovered that the GCH1/BH_4_ cascade was also important in protecting neuronal and hSOD1^G93A^ models of amyotrophic lateral sclerosis (ALS) from ferroptosis [266]. In that paper, it was demonstrated that the protein speedy/RINGO cell cycle regulator family member A (SPY1), whose expression was downregulated during ferroptosis in ALS models, can inhibit ferroptosis by regulating the GCH1/BH_4_ axis and transferrin receptor protein 1 (TFR1). Further results obtained in vivo proved that SPY1-recombinant virus injected into the lateral ventricle of hSOD1^G93A^ mice prolonged survival and reduced TFR1 levels. These results are important for new therapy tools to counteract the build-up of lipid reactive oxygen species in ALS. In summary, the GCH1/BH_4_-dependent pathway can capture lipid radicals and overcome ferroptosis.

##### Microsomal Glutathione Transferase 1 (MGST1) (Inactivation of ALOX5 and/or Autophagy)

A particular mechanism of ferroptosis inhibition is mediated by the microsomal glutathione transferase 1 (MGST1) [267]. Glutathione transferases (GSTs) are metabolic enzymes involved in the detoxification of ROS/RNS, xenobiotics, and electrophile molecules. GST-mediated activity implicates the conjugation of GSH to a wide range of substrates; GSTs comprise two enzyme superfamilies, one microsomal and the other cytosolic. The MGST1 enzyme is a membrane-bound enzyme that in addition to glutathione transferase activity also possesses a peroxidase activity; hence, it can manage not only electrophile chemicals and drugs but also products of peroxidation, thus protecting cells under oxidative stress conditions [268,269]. MGST1 was found upregulated during ferroptosis induced by erastin or RSL3 treatments in pancreatic cancer cells. Notably, MGST1-mediated protection involves the interaction and partial inactivation of the lipid peroxide-producing enzyme ALOX5 [267], whereas other lipoxygenases like (A)LOX12 or (A)LOX15 do not interact with MGST1, at least in pancreatic cancer cells. Inhibition of ALOX5 by MGST1 appeared crucial in resistance because MGST1 knockdown further increased the production of oxidized PUFA-PEs. The authors found that the upregulation of MGST1 gene transcription during ferroptosis was dependent on NRF2 and that knockdown of either NRF2 or MGST1 can increase cell sensitivity to ferroptosis inducers. Other results obtained in an in vivo xenograft model confirmed that loss of NRF2 or MGST1 can sensitize pancreatic cancer cell xenografts to ferroptosis [267]. In SGC7901 gastric carcinoma cells, MGST1 expression was downregulated in ferroptosis induced by erastin, sorafenib, and RSL3, while its ectopic expression lowered ferroptosis coupled with the inactivation of autophagy mediated by the Akt/GSK-3β pathway [270]. Various reports have indeed proved that autophagy is an upstream event, triggering ferroptosis through increasing intracellular iron levels with consequent ROS generation (reviewed in [29,271,272]).

##### AKR1C Family Members (Scavenging of HNE)

Aldo-keto reductase family 1 (AKR1) consists of metabolic enzymes with a NAD(P) H-dependent activity that catalyze the conversion of aldehydes/ketones to their corresponding alcohols, and in humans, mostly participate in steroid hormone metabolism [273]. Of these, the AKR1C1 family members can mediate resistance to ferroptosis, being also involved in the detoxification of lipid breakdown products, like HNE [274]. In this context, increased expression of AKR1C members was first reported by Dixon et al., who found that DU145-derived cell clones were resistant to cysteine depletion and displayed strong upregulation of AKR1C1 -3 mRNAs, thus suggesting that AKR1C1 isoforms can possibly prevent ferroptosis [48]. Furthermore, in melanoma cell lines that potentially can induce the ferroptotic process, AKRC1s are implicated in resistance to ferroptosis execution by enhancing the detoxification of reactive aldehydes during ferroptosis. Accordingly, inhibition of AKRC1s resensitizes melanoma cells to ferroptosis execution [275]. Another study confirmed the protective role of AKR1C1 in corneal epithelial cells against lipid peroxidative stress through NRF2 transcriptional activation and demonstrated that AKR1C1 inactivation pushes ferroptosis [276].

### 3.3. Other Protective Mechanisms (Composition of Phospholipid Membranes)

#### 3.3.1. Stearoyl-CoA Desaturase 1 (SCD1)

SCD1 is an iron-dependent endoplasmic reticulum enzyme that converts saturated fatty acids into Δ^9^-monounsaturated fatty acids and has been reported to be protective against ferroptosis in cancer cells [277,278]. A recent work revealed that oncogenic activation of the PI3K-AKT-mTOR, frequent in human cancer, is causally related to ferroptosis resistance primarily sustained by lipogenesis mediated by mTORC1 activation with consequent induction of sterol regulatory element-binding protein 1 (SREBP1), a transcription factor that in turn increases SCD1 expression, in part by increased NRF2 [279].

#### 3.3.2. Acyl-CoA Synthetase Long Chain Family Member 3 (ACSL3)

ACSL3, like ACSL4, converts fatty acids into fatty acyl-CoAs for esterification into membrane phospholipids and is considered a protective enzyme in ferroptosis [147,280]. Its beneficial effect was assessed by examining the protective role of MUFAs in ferroptosis by the Dixon group [147]. The authors discovered that in cells treated with oleic acid, ACSL3 is specifically required to counteract ferroptosis induced by erastin. The ACSL3 activity produces MUFA-CoAs that directly compete with PUFA-CoAs for incorporation into lysophospholipids during plasma membrane remodeling [147]. A recent study showed that in gastric cancer, ACSL3 acts downstream of methionine adenosyltransferase 2α (MAT2A), which produces S-adenosylmethionine (SAM), necessary not only for the transsulfuration pathway but also the transmethylation pathway and polyamine synthesis [281]. In this context, MAT2A indeed mediates ACSL3 upregulation by increasing the trimethylation of lysine-4 on histone H3 (H3K4me3) on the ACSL3 promoter, thus favoring ferroptosis resistance [282].

#### 3.3.3. MBOAT1 and MBOAT2/LPCAT4

Liang et al. very recently unveiled that two acyltransferase enzymes, named MBOAT1/2 (membrane bound O-acyltransferase domain containing 1/2) can inhibit ferroptosis by remodeling the membrane phospholipid profile and this protective effect is independent of GPX4 or FSP1 activity [160]. This acyltransferase indeed catalyzes the selective transfer of MUFAs into lyso-phosphatidylethanolamine (lyso-PE), which increases the PE-MUFA pool, thus potentially reducing PE-PUFA, and this consequently can restrain ferroptosis. In this study, MBOAT2 was first identified through a whole-genome CRISPR activation screen as a ferroptosis-suppressing gene, then the authors tested the effects of the other member, MBOAT1, and found that it also suppresses ferroptosis by sharing a similar mechanism. Overexpression of MBOAT2 significantly inhibited RSL3 or GSH depletion-induced ferroptosis in human fibrosarcoma cells, even in the absence of GPX4 and FSP1. Moreover, this work revealed that MBOAT2 and MBOAT1 are transcriptionally induced by sex hormone receptors, like androgen receptor (AR) and estrogen receptor (ER), respectively. This implicates a diversified role of the two members in biological contexts and suggests that specific receptor inhibition by downregulating MBOAT expression could sensitize prostate cancer and breast cancer cells to ferroptosis [160].

## 4. Antioxidant Systems and Transcription Factors

Although the molecular mechanisms and/or metabolic pathways governing ferroptosis are still poorly understood, this process seems to be under the control of numerous transcription factors (TFs) [14,15], with most of them acting as transcriptional modulators of specific genes involved in redox homeostasis and lipid peroxidation processes, as well as in iron/heme metabolism and inflammation. Many TFs work by reinforcing the antioxidant systems, including the GPX4/glutathione/cysteine pathway, mevalonate pathway, and ferroptosis inhibitory protein 1 (FSP1)–coenzyme Q_10_ (CoQ_10_) pathway, therefore preventing ferroptosis, whereas others foster ROS/RNS production, lipid oxidation, and iron accumulation, therefore pushing ferroptosis [73] (Figure 6).

### 4.1. NRF2 Pathway

NRF2 is one of the most important transcription factors in ferroptosis sensitivity that exerts an anti-ferroptotic effect by modulating the expression of many genes involved in antioxidant response and iron metabolism as well as in lipid homeostasis (reviewed in [283]). NRF2 function is mainly regulated by the Kelch-like ECH-associated protein 1 (Keap1). Keap1 plays a crucial role in basal conditions to promote NRF2 ubiquitination and proteasome degradation, thus keeping NRF2 at low levels. Under stressed conditions, Keap1 is inactivated and consequently de novo synthesized NRF2 can move into the nucleus, where it forms heterodimers mainly with the small MAF (musculoaponeurotic fibrosarcoma, MafF, MafK, and MafG) proteins to bind the antioxidant response elements (AREs) present in its target gene promoters, activating their transcription. Nrf2 can also form heterodimers with other transcription factors, thus extending the repertoire of its gene targets. Keap1 inactivation results in NRF2 stabilization and consequent upregulation of NRF2 target genes [284], a condition that favors resistance against ferroptosis, as demonstrated by Cao et al. through genome-wide human haploid cell genetic screening [285]. Importantly, although the dominant regulation of NRF2 occurs at the protein level, NRF2 expression is also modulated at the transcriptional level by different oncogenes, such as KRAS, BRAF, and c-MYC [286], and thus many cancer cells display constitutively activated NRF2 [287]. Furthermore, NRF2 enhances the resistance of cancer cells to chemotherapeutic drugs. Accordingly, loss of NRF2 confers increased sensitivity to electrophilic, xenobiotic, metabolic, and ferroptosis inducers in cancer cell lines [288].

A pioneering study in 2016 demonstrated that during treatment with various ferroptosis inducers (e.g., erastin, BSO, and sorafenib), the NRF2 protein levels were increased in hepatocarcinoma cell lines [289]. This was caused by the inactivation of Keap1 through the p62/sequestosome1 pathway (allowing NRF2 stabilization [284]). It was demonstrated that NRF2-dependent transcriptional activation of genes involved in heme (i.e., HO-1), iron (i.e., FTH1), and ROS metabolism (i.e., NQO1) could protect against ferroptosis and that NRF2 inhibition potentiated the sensitivity to ferroptosis in vitro and in tumor xenograft models [289]. In general, NRF2 can control ferroptosis cascades by modulating the expression of multiple genes involved in the regulation of iron levels, in the synthesis of GSH, and in the removal of lipid hydroperoxides, as well as those of the FSP1-dependent CoQ system [290]. NRF2 buffers the labile iron pool by upregulating ferritin gene expression, which favors iron storage and modulates the expression of ferroportin, the only known exporter of iron in mammals, therefore protecting against iron overload-induced oxidative stress and by inducing metallothionein 1G expression, which plays an essential role in the detoxification of divalent metal ions [291]. A recent work demonstrated that NRF2 controls the synthesis and degradation of ferritin by directly inducing the expression of the HERC2 gene, containing functional AREs in its promoter and coding for an E3 ubiquitin ligase, and by indirectly regulating autophagic protein VAMP8 levels [292]. In that paper, the authors demonstrated that NRF2 can contribute to iron homeostasis through three mechanisms: (a) ferritin synthesis through the NRF2–HERC2–FBXL5–IRP1/2 axis; (b) ferritin degradation via ferritinophagy through the NRF2–TFEB–VAMP8–cascade, and (c) ferritin recruitment to the autophagosome via the NRF2–HERC2–NCOA4 axis. Indeed, ablation of NRF2 increases apoferritin in the autophagosome, promotes LIP accumulation, and consequently enhances sensitivity to ferroptosis in human ovarian cancer cells and preclinical models [292]. In addition, NRF2 controls iron levels through regulation of heme metabolism-related genes, including (i) HO1, which degrades heme, (ii) ABCB6, which transports heme/porphyrin across intra- and extra-cellular membranes; (iii) FECH (ferrochelatase), which inserts iron into protoporphyrin IX, (iv) SLC48A1, which carries heme from lysosomes to cytoplasm, and (v) BLVRA/B (biliverdin reductase), which catalyzes the second step in heme degradation (biliverdin reduction to bilirubin). NRF2 intervenes in lipid protection by regulating a set of genes involved in different aspects of lipid metabolism. Firstly, NRF2 can directly regulate the GPX4 protein content via binding to the ARE element present in its promoter [293] or indirectly by increasing the expression of ARE-target genes (containing promoter ARE sequences) coding for enzymes that provide endogenous thiol resources, including GSH-synthesizing and -regenerating enzymes (GCLC/GCLM, GSS, and GSR), cystine transporters (SLC7A11), and enzymes synthesizing cysteine, as well as genes encoding components of antioxidant systems like TRX and TXNRD1 and metabolic enzymes [glucose-6·phosphate dehydrogenase (G6PD), 6-phosphogluconate dehydrogenase (PGD), isocitrate dehydrogenase 1 (IDH1), and malic enzyme 1 (ME1)] that produce NADPH, a cofactor necessary to reconstitute reduced GSH, TRXs, and PRDXs from their disulfide forms, thus ensuring the reduced state of cysteine thiols and FSP1 activity. Other genes regulated by NRF2 are the AKR1C family members and MGST1, which contribute to the detoxification of lipid-derived reactive aldehydes and lipid peroxides, as well as FSP1, producing reduced coenzyme Q_10_ that traps lipid peroxides. Concerning lipid content, NRF2 regulates the expression of genes involved in lipid metabolism by down-modulating the expression of genes involved in lipogenesis and up-regulating genes involved in lipolysis as well as genes involved in lipid transport and uptake, thus influencing ferroptosis susceptibility through controlling lipid composition. Recently, Takahashi et al., by using three-dimensional (3D) cancer spheroid models from lung cancer cell lines (A549 and H1347 bearing hyperactivated NRF2), demonstrated that NRF2 prevents lipid peroxidation through reducing ROS levels in inner spheroid cells, which are more susceptible to ferroptosis, as proven by the same authors [294]. CRISPR-Cas9 screens identified, along with NRF2 and MAFG, three NRF2-dependent genes of the glycolytic/pentose phosphate pathway, as well as GPX4, as common hits required for growth whereas NRF2 silencing renders spheroid cells highly vulnerable to ferroptosis, in spite of a compensatory GPX4 increase, indicating the predominant role of NRF2 in shielding against redox stress [294]. NRF2 can also form heterodimers with members of the AP-1 family, like c-JUN, or ATF members, acting on ARE-target genes [295,296]. More recently, Zhang et al. demonstrated in lung cancer that the induction via NRF2/c-JUN-CBS expression of the transsulfuration pathway is important in the protection against erastin- or RSL3-induced ferroptosis [297]. This pathway acts through the transcription factor nuclear receptor subfamily 0 group B member 1 (NR0B1, also known as DAX1 dosage-sensitive sex reversal-AHC critical region on the X-chromosome gene 1) to push cysteine production. Indeed, manipulation of NR0B1 levels affected ferroptosis by influencing GSH, ROS, MDA, and iron levels. RNA-seq analysis in A549 lung cells harboring knocked-down or overexpressed NR0B1 demonstrated that NR0B1 expression positively correlated with NRF2, c-JUN, and CBS by directly binding to the promoters of NRF2 and c-JUN, which in turn form functional dimers that promote CBS expression. Implantation in mice of lung cancer cells A549 harboring depleted NR0B1 further proved that tumor growth was restrained in the absence of NR0B1 and favored RSL3-induced ferroptosis [297]. However, NRF2 signaling could be highly context-dependent; for example, GPX4 could be either upregulated or downregulated by NRF2 in different cancer cell lines [260], making NRF2 a multifaceted modulator of the anti-ferroptotic response [298]. It is important to underline that most of the above genes can also be transcriptionally regulated by different transcription factors. For example, SLC7A11 can be induced by ATF4 (activating transcription factor 4) and inhibited by the tumor suppressor p53 or ATF3 (activating transcription factor 3) under basal conditions [299].

### 4.2. ATF Signaling

Activating transcription factor 2 (ATF2/CREB2) is a multi-faceted leucine zipper transcription factor that binds to the cAMP responsive element (CRE). Generally, in response to many cell stress conditions, preferentially oxidative stress, ATF2 is phosphorylated by different kinases, including JNK and p38/MAPK. Then, phosphorylated ATF2 translocates to the nucleus where, in combination with different interactors such as Fos, c-Jun, CREB, and ATF1, it activates its gene targets [300]. Numerous evidence indicates that ATF2 can act either as an oncogene or antioncogene in different cancer types depending on its expression level and/or subcellular localization. A recent study of Xin Xu et al. reported that ATF2 is elevated in gastric cancer (GC) [301]. Further experiments in GC cells demonstrated that ATF2 knockdown inhibited malignant phenotypes and potentiated ferroptosis induced by sorafenib, a protein kinase inhibitor drug, causing endoplasmic reticulum stress, GSH depletion, and iron-dependent lipid radical accumulation, while overexpression of ATF2 reduced sorafenib-induced ferroptosis sensitivity. Specifically, ATF2 protects GC cells from sorafenib-mediated ferroptosis by raising the expression of the heat shock protein-110 (HSPH1, also called HSP105 or HSP110), which physically interacts with SLC7A11 and prevents its degradation. Indeed, ATF2 can bind the HSPH1 promoter region, consequently activating its transcription, whereas HSPH1 is downregulated following ATF2 knockdown. HSPH1 silencing in gastric adenocarcinoma AGS cells stably overexpressing ATF2 showed downmodulation of SLC7A11 at the protein but not at the mRNA level, thus highlighting that HSPH1 can interact with and improve SLC7A11 stability in GC cells, thus contributing to the uptake of cystine required for GSH synthesis [301]. Furthermore, ATF2 activated through the JNK1/2 pathway plays a role in attenuating ferroptosis induced by treatments with BETi (inhibitors of bromo- and extra-terminal domain) in cancer cells and this was mediated by increased expression of NRF2 [302]. In fact, while depletion of ATF2 increased the ferroptosis induced by BETi treatment in MB-231 cells, ATF2 modulation does not affect BETi-induced ferroptosis in NRF2-silenced cells [302]. It was also demonstrated that ATF2 plays a crucial role in maintaining GSH homeostasis in human mesenchymal stem cells (MSCs) by modulating the CREB1–NRF2 redox homeostasis signaling pathway [303]. In MSC cells, ascorbic acid 2-glucoside (AA2G) treatment increased the mRNA levels of NRF2 and several genes related to GSH synthesis (GCLC and GCLM) and redox cycling (GSR and PRDX1) previously reported as targets of the CREB1–NRF2 pathway. ATF2 knockdown impaired the AA2G-mediated induction of the GSH-related genes targeted via the CREB1–NRF2 pathway and counteracted the nuclear translocation of NRF2 protein, suggesting an interplay between the ATF2 and CREB1–NRF2 signaling cascades in MSCs [303].

ATF3 is a stress-inducible transcription factor and a member of the activation transcription factor/cAMP responsive element-binding (CREB) protein family. It is expressed at low levels in normal conditions but it can be upregulated by a variety of stress signals, including hypoxia, cytokines, DNA damage, oxidative stress, endoplasmic reticulum stress, and cell injury. A growing body of evidence shows that ATF3 modulates anti-inflammatory and immune response, oncogenesis, and metabolic homeostasis [304,305]. ATF3 regulates gene expression by binding to consensus ATF/CREB cis-regulatory elements in DNA targets via a basic-region leucine-zipper (bZIP) domain. In addition, ATF3 can interact with its family members and other proteins, such as p53 and Tip60, and it regulates cellular functions independently of its transcriptional activity. Regarding ferroptosis, ATF3 can suppress system Xc- and predispose cells to ferroptosis by repressing SLC7A11 expression. 

In detail, Liyuan Wang et al. demonstrated that ATF3 over-expression sensitized human fibrosarcoma HT1080 cells to erastin-induced ferroptosis death by causing an excessive increase of lipid peroxide levels. Conversely, ATF3 knockout dramatically suppressed erastin-induced ferroptosis and lipid peroxidation with a negligible effect on the amount of released glutamate and decreased intracellular GSH levels. By genome-wide ChIP-seq analysis and gene expression analysis, the authors revealed that ATF3 binds the SLC7A11 promoter region proximal to the transcription start site and represses its expression. Interestingly, lentiviral-mediated SLC7A11 expression in ATF3 over-expressing fibrosarcoma cells restored system Xc- function and impaired ATF3-mediated ferroptosis. Thus, ATF3 induces ferroptosis by suppressing system Xc- through SLC7A11 expression downregulation [306].

Furthermore, Yilan Li et al. showed that ATF3 also promotes sorafenib-induced ferroptosis by suppressing SLC7A11 in cardiomyoblasts. In fact, ATF3 was induced upon sorafenib treatment and, notably, knockdown of ATF3 failed to restrain SLC7A11 expression, decreased cellular ROS and lipid peroxides, and preserved the ΔΨm in sorafenib-treated H9c2 cardiomyoblasts [307]. Xi Chen et al. recently demonstrated that ATF3 activation mediated by sirtuin1 (SIRT1) deacetylase decreased SLC7A11 and GPX4 expression in glioma cells treated with the GPX4 inhibitor RSL3, causing ferroptosis, associated with intracellular ferrous iron increase and lipid peroxidation [308]. They demonstrated that manipulation of SIRT1 activity either through activator (SRT2183) or inhibitor (EX527) treatment, or decreasing SIRT1 levels by siRNA, impacted RSL3-triggered SLC7A11 and GPX4 expression; in line with this, ATF3 siRNA impaired the RSL3-SLC7A11 and GPX4 downregulation of expression and reduced the sensitivity to RSL3-induced ferroptosis [308].

ATF4 is another bZIP stress-responsive transcription factor of the ATF/CREB family that binds to conserved cAMP-responsive element (CRE) to modulate the transcription of target genes. ATF4 expression is activated in response to numerous stress signals such as amino acid starvation, anoxia/hypoxia, and endoplasmic reticulum stress [309]. Albeit ATF4 could play a dual role in favoring or exacerbating cell death, many studies have demonstrated a protective role of ATF4 in ferroptosis as a compensatory factor for SLC7A11 expression. In pancreatic ductal carcinoma, ATF4 has been demonstrated to have a protective effect in ferroptosis by transcriptional activation of SLC7A11 and by maintaining GPX4 stability through induction of the heat shock protein family A (Hsp70) member 5 (HSPA5), an ER-associated molecular chaperone [310]. In mouse models and melanoma cells deficient in xCT, induction of ATF4 and NRF2 under conditions of cystine- or GSH-depletion enabled a compensatory response by increasing cysteine synthesis via the transsulfuration pathway and the antioxidant system via the thioredoxin pathways [311]. In the hepatocarcinoma context, ATF4 activation sustained by the YAP/TAZ pathway induced SLC7A11 to protect against sorafenib-induced ferroptosis [312]. More recently, He et al. demonstrated that young MUP-uPA mice (undergoing hepatic ER stress under a high-fat diet (HDF)) lacking hepatocyte ATF4 expression exhibited enhanced liver injury and disruption of redox homeostasis associated with lower intracellular GSH and decreased mRNA levels of SLC7A11, SLC7A5, and superoxide dismutase 2 (SOD2), sensitizing cells to ferroptosis [313]. In fact, ATF4 ablated-cultured hepatocytes are more sensitive to RSL3 treatment, and in vitro reconstitution of either ATF4 or SLC7A11 protected ATF4-deleted hepatocytes against RSL3-mediated ferroptosis. Luciferase assays demonstrated that ATF4 stimulates SLC7A11 expression by cooperating with NRF2. Indeed, ATF4 does not induce SLC/A11 in the absence of NRF2 [313]. Finally, liver-specific ATF4 deletion provoked lipid peroxidation and hepatic damage, and accelerated liver cancer in HFD-fed MUP-uPA mice that could be counteracted by forced SLC/A11 expression [313]. Therefore, ATF4 may interact with NRF2 to transcriptionally activate multiple protective factors against ferroptosis, including SLC7A11. Indeed, additional experiments revealed that ATF4-dependent tumor-promoting effects are mediated by transcriptional targeting the glutamate antiporter xCT/SLC7A11, and the knockdown of ATF4 increases the ferroptosis cell death induced by different ferroptosis agents, including sorafenib, erastin, and RSL3, associated with lipid peroxidation accumulation [299]. In other studies, ATF4 expression was found to sustain the aggressiveness of primary brain tumors by enhancing growth, angiogenesis, migration, and anchorage-independent cell growth and also by conferring a multidrug-resistance phenotype, whereas ATF4 knockdown reduced the malignancy features in human glioma cells [299].

### 4.3. TFAP2 Pathway

Another family of transcription factors involved in the attenuation of ferroptosis is related to the transcription factor AP-2, comprising five members (TFAP2A–E) [314]. The transcription factor alpha (TFAP2A) has been linked to NRF2 signaling. Indeed, TFAP2A silencing downmodulates in vitro the expression of NRF2 and NRF2-dependent genes linked to ferroptosis, including HO-1, FTH1, and NQO1 [315]. TFAP2 gamma (TFAP2C) also prevents ferroptosis by regulating the transcription of various genes under different conditions. Ishraq Alim et al. demonstrated that TFAP2, activated in neurons by supraphysiological levels of selenium (Se), together with SP1 achieved antioxidant GPX4 expression to protect neurons from ferroptosis and improved behavior in a hemorrhagic stroke model. In particular, ChIP assays showed that neurons exposed to Se significantly increased the TFAP2 and SP1 occupancy on the Se-responsive region of GPX4, which binds the same or adjacent DNA binding sites. In addition, overexpression of TFAP2 or SP1 regulated the mRNA expression of several other genes induced by ferroptosis stimuli, such as GPX3, TXNRD1, SELK, and MSRB1, and protected neurons from hemin or HCA-induced ferroptosis [316]. 

Recently, Xingkang Jiang et al. reported that TFAP2 activates the transcription of the prostate cancer-associated transcript 1 (PCAT1), an oncogenic long noncoding RNA [317]. PCAT1 was upregulated in docetaxel (DTX)-resistant PCa prostate cancer cells and in prostate cancer (PCa) patients, whereas PCa patients with higher serum PCAT1 levels had an unsatisfactory response to DTX chemotherapy. Besides, PCAT1 up-regulation decreased erastin- and DTX-induced cytotoxicity in PCa cells. Conversely, prostate cancer cells with PCAT1 knockdown developed tolerance toward ferroptosis induced by erastin or DTX treatments, reducing the lipid ROS content and iron overload. Investigation of the molecular mechanism of TFAP2C-mediated ferroptosis resistance demonstrated that TFAP2C knockdown efficiently inhibited the transcription of PCAT1 and SLC7A11. Since TFAP2C is able to bind only the PCAT1 promoter, the decrease of SLC7A11 expression is PCAT1-mediated. In fact, the authors observed a downmodulation of SLC7A11 mRNA and protein following PCAT1 knockdown and, also, showed that PCAT1 silencing decreased SLC7A11 expression in xenograft mouse models. Moreover, RNA pull-down and immunoprecipitation assays revealed that PCAT1 interacts with c-Myc protein, increasing its stability and consequently c-Myc activates SLC7A11 transcription by binding to its promoter. Finally, PCAT1 also positively regulates SLC7A11 expression by interacting with and sponging miR-25-3p, which can target the 3′UTR of SLC7A11, as proven by luciferase assays. Thus, PCAT1, activated by TFAP2C, in turn, hinders ferroptosis cell death through regulation of the c-Myc/miR-25-3p/SLC7A11 signaling pathway [317].

### 4.4. JAK-STAT Pathway

Multiple studies have indicated that alteration of redox circuits and increased lipid peroxidation can activate inflammatory pathways eliciting pro-inflammatory cytokines that, in turn, aggravate oxidative stress leading to excessive lipid peroxidation [318]. Janus kinase-signal transducer and activator of transcription (JAK-STAT), nuclear factor-κB (NF-κB), inflammasome, cGAS-STING, and MAPK signaling pathways are indeed key regulators of ferroptosis.

JAK/STAT signaling comprises membrane receptors that after binding with ligands and dimerization, recruit constitutively associated non-receptor kinases called Janus kinases (JAKs), which phosphorylate the receptors, yielding docking sites for cytosolic STAT transcription factors (STAT1-4, STAT5a, STAT5b, and STAT6) that also become phosphorylated and activated. Phosphorylated STATs (pSTATs) bind to each other, forming homo- or hetero-dimers that translocate into the nucleus to mediate gene transcription by binding to specific promoter regions (reviewed in [319]). The JAK/STAT pathway mediates signal transduction for numerous growth factors/hormones and cytokines (>50 ligands), in particular interferons (IFNs) and interleukins (ILs) regulating differentiation, proliferation, apoptosis, and cell survival, depending on the specific signals and cellular context and driving inflammatory diseases, lymphomas, leukemias, and different solid tumors [320,321]. In addition, other signaling pathway members and/or other regulatory circuits can regulate JAK/STAT signaling by pushing or inhibiting STAT phosphorylation and translocation [321]. Increasing data indicate that STATs can be oncogenes or tumor suppressor factors, and any treatment based on STAT modulators should take the multifaced function of these transcription factors in different tumors into account [322].

The JAK/STAT pathway activated by IFNγ, which binds two receptor subtypes, IFN-γ receptor (IFNGR)-1 and-2, participates in ferroptosis. The work of Tsoi et al. reported that IFNγ under erastin treatment has a role in potentiating ferroptosis of human melanoma cells through STAT1 increases [323]. Next, Wang et al. demonstrated in co-cultured cell systems that IFNγ (released by effector CD8+ T cells) sensitizes tumor cells to ferroptosis induced by RSL3 or erastin by reducing Xc- system activity, consequently depleting GSH and increasing lipid peroxidation/ferroptosis of tumor cells [324]. In this context, IFN-γ directed the binding of STAT1 to the SLC7A11 promoter, as demonstrated by ChIP assays. In line with this, STAT1 deficiency abolished the IFN-γ contribution and cyst(e)ine depletion by cyst(e)inase (an engineered enzyme that breaks down extracellular cysteine/cystine) increases T cell-mediated ferroptosis [324].

Furthermore, IFNγ enhanced erastin-induced ferroptosis, increasing lipid peroxidation, iron, mitochondrial damage, and ROS in adrenocortical carcinoma cells [325]. In particular, Western blot, RT-qPCR, and immunofluorescence analyses showed that IFNγ, through STAT1/STAT3 and interferon regulatory factor 1 (IRF1) activation, enhanced erastin-induced ferroptosis associated with repression of SLC7A11 expression. Accordingly, IFNγ treatments failed to increase lipid peroxidation and mitochondrial damage in adrenocortical carcinoma cells with SLC7A11 over-expressed or STAT1 inhibited [325]. Another work in hepatocellular carcinoma cells demonstrated that IFNγ treatment sensitized Bel7402 and HepG2 cells to ferroptosis inducers (erastin or RSL3) by reducing the GSH content and increasing lipid peroxidation mediated by the JAK/STAT pathway that reduced mRNA/protein expression of SLC3A2 and SLC7A11, consequently inhibiting system Xc- activity, provoking ROS increase and ΔΨm decrease. Of note, IFN-γ treatment in vivo abated the growth of xenograft tumors through increasing lipid oxidation [326].

Differently from the cancer context, age-related macular degeneration (AMD) causes cell death in retinal pigment epithelium (RPE) that can be coupled to IFN-γ increases. The work of Wei et al. demonstrated indeed that IFN-γ promoted ferroptosis in human retinal pigment epithelial cells (ARPE-19) and this was accompanied by iron and ROS increase, lipid peroxidation, and GSH depletion [327]. These effects were linked to inhibition of SLC40A1 expression (which exports iron), and to activation of the JAK1/2-STAT1 pathway that lowered SLC7A11/GPx4 and GSH levels, as demonstrated with specific inhibitors. These results were further confirmed in vivo by using IFN-γ-treated mice as well as AMD models generated by sodium iodate treatment that provokes retinal degeneration, suggesting that inhibition of ferroptosis (and IFN-γ) could be a possible target for AMD [327].

In a previous study, Gao et al. demonstrated that STAT3 is a positive regulator of ferroptosis through activation of lysosomal-dependent cell death in human pancreatic ductal adenocarcinoma (PDAC) cell lines. Pharmacological or siRNA inhibition of STAT3 interferes with erastin-induced ferroptosis. Indeed, lysosomal dysfunction induced by selective cathepsin inhibition or by vacuolar-type H+-ATPase hampering restricted erastin-induced ferroptosis in PDAC cells. Finally, the authors displayed that STAT3-mediated cathepsin B (CTSB) expression was required for ferroptosis [328].

On the contrary, Shumin Ouyang et al. established a negative STAT3–ferroptosis regulatory axis in gastric cancer. In particular, the authors demonstrated by ChIP-qPCR analysis that STAT3 binds to the promoter regions of three ferroptotic genes, namely GPX4, SLC7A11, and FTH1, to positively regulate their expression [329]. STAT3 knockdown in gastric cancer cells provoked a significant inhibition of GPX4, SLC7A11, and FTH1 at both the mRNA and protein levels, and also increased lipid peroxidation and ROS, and enhanced intracellular Fe^2+^, thus reducing the GSH/GSSG ratio in gastric cancer cells. A similar effect was observed in gastric cancer cells treated with the STAT3 inhibitor W1131 caused by reduced STAT3 occupancy on the GPX4, SLC7A11, and FTH1 promoters. Interestingly, the combination of W1131 and 5-FU (5-fluorouracil) re-sensitizes chemo-resistant cancer cells to 5-FU. Thus, these data underlie the importance of the use of JAK/STAT inhibitor drugs to re-establish the sensitivity of cells to ferroptosis, thus increasing the chemotherapy response [329].

Other evidence that STAT3 inhibition may induce ferroptosis comes from the work of Liu et al. [330]. They reported that cisplatin-resistant osteosarcoma MG63 and Saos-2 cells have elevated pSTAT3 and NRF2 levels. Treatment with erastin and RSL3 decreased the viability of cisplatin-resistant cells accompanied by increased levels of ROS, lipid peroxides, MDA, and decreased GPx4 protein levels. In addition, treatments with BP-1-102, a STAT3 inhibitor, abolished resistance to cisplatin, indicating that the STAT3/NRF2/GPx4 signal has a critical function in the drug resistance of osteosarcoma cells [330]. Also, Yi Luo et al. found that the flavonoid bavachin inhibits the viability of MG63 and HOS osteosarcoma cell lines by inducing ferroptosis through the STAT3/p53/SLC7A11 axis [331]. The authors showed that bavachin treatment down-modulated GPX4 and SLC7A11 expression, increased p53 protein levels, and inhibited STAT3 activation. It was also demonstrated that bavachin increased the intracellular ferrous iron concentration and the ROS, and MDA levels, consequent to SLC7A11 downregulation and p53 upregulation, this latter being mediated by STAT3 inactivation. Accordingly, SLC7A11 overexpression decreased the features of bavachin-induced ferroptosis, and the use of PFT-α, a p53 inhibitor, upregulated SLC7A11, thus attenuating the bavachin-induced ferroptosis. Finally, osteosarcoma cells overexpressing STAT3 failed to upregulate p53 expression in response to bavachin and no longer induced ferroptosis. Therefore, STAT3 inhibition by bavachin caused p53 upregulation that inactivated SLC7A11, thus inducing ferroptosis [331]. These data further extend previous work in human breast cancer cells demonstrating that the JAK/STAT pathway, specifically STAT3 and STAT5 proteins, negatively regulate xCT expression [332].

### 4.5. NF-κB Pathway

Nuclear factor-κB (NF-κB) represents a family of transcription factors whose activity is induced by several cytokines, such as tumor necrosis factor-alpha (TNF-α), interleukin-1 beta (IL-1β), ROS/RNS, and harmful stimuli [333]. NF-κB controls crucial physiological processes, such as immunity [334], inflammatory response [335], proliferation, cell death, cancer, and nervous system function [336] by regulating various pro-inflammatory genes (cytokines and chemokines), immunoreceptors, cell adhesion molecules, and several cell cycle regulators [cyclin A, cyclin D1, or cyclin-dependent kinase 6 (CDK6)]. NF-κB activity has a crucial role in the development of malignant tumors. Its constitutive activation enhances cancer cell proliferation and promotes the inhibition of apoptosis and chemotherapy resistance in many cancer types [337].

The NF-κB family includes five proteins: Rel (cRel), p65 (RelA), RelB, p105, and p100. The last two proteins undergo proteolysis, producing the p50 (NFκB1), and p52 (NFκB2) subunits. Among these, p65, cRel, and RelB proteins contain an N-terminal Rel-homologous domain (RHD), necessary for DNA binding and dimerization with other family members, and a C-terminal transactivation domain (TAD), associated with transcriptional activation. p50 and p52 proteins have only an RHD, but not a TAD domain. The homo- and hetero-dimers of NF-κBs (i.e., RelB-p50, RelB-p52, and p50-p65) bind specific consensus sequences on target genes for transcription regulation [337]. Normally, NF-κB factors are in the cytoplasm where they associate with members of the inhibitor family (IκBα, β, and ε) that prevent nuclear translocation. The NF-κB/IκB association is impaired by phosphorylation of IκBs, mediated by a specific IKK (IκB kinase) complex, that causes IκB ubiquitination and subsequent proteasomal degradation, thus liberating NF-κB, which translocates into the nucleus to modulate downstream gene transcription [338].

NF-κB plays a dual role in ferroptosis events; in fact, it can either activate or inhibit ferroptosis, depending on the tissue/tumor type and on the specific stimulus. It can indeed modulate ferroptosis signaling by directly regulating several genes associated with iron accumulation and/or lipid peroxidation, as recently reviewed in [318]. Of note, a crosstalk between NF-κB/p65 and NRF2 was reported. In particular, NF-κB/p65 could counteract the NRF2/ARE cascade, which consequently may influence ferroptosis sensitivity. In fact, the knockdown of p65 by siRNA enhanced the NRF2-mediated activation of HO-1, GPX2, and NQO1 genes in hepatoma HepG2 cells, whereas co-transfection of HO-1(ARE), GPX2(ARE), or NQO1(ARE)-Luc reporters with either NF-κB, p65, or IKKβ expressing vectors inhibited ARE-driven expression [339]. In p65 overexpressing cells, co-immunoprecipitation experiments demonstrated that p65 inhibits NRF2 transcriptional activity through sequestering CBP (CREB binding protein), a common coactivator of p65 and NRF2. Because NRF2 cannot bind to CBP, its transcriptional activity was inhibited. Besides, p65 fosters the association of HDAC3 with either MafK or CBP, favoring recruitment of HDAC3 to ARE sequences and transcriptional repression [339].

A recent study by Xu et al. reported that the NF-κB pathway exerted a dominant protective function in ferroptosis occurring during ulcerative colitis (UC), a phenomenon linked to endoplasmic reticulum (ER) stress. In fact, a reduction of ER stress lowered ferroptosis in intestinal epithelial cells and ameliorated colitis. The authors noted that epithelial and mesenchymal cells in the colonic sections from UC patients presented elevated levels of activated p65 and mice models with targeted p65 deletion exacerbated colitis induced by dextran sulfate sodium. As expected, phosphorylation of p65 blunted ER stress-mediated ferroptosis, and this was mediated by the direct interaction of p65 with eIF2α [340]. In line with NF-κB’s ferroptotic protective function, another study reported that BAY 11-7085, a selective inhibitor of the NF-κB pathway, could induce this type of cell death through upregulating the NRF2-SLC7A11-HO-1 signaling pathway [27]. Detailed analyses in human breast cancer cells demonstrated that BAY mediated NRF2 nuclear translocation and increased HO-1 expression while cells deficient in SLC7A11 presented exacerbated ferroptosis. HO-1 also elicited ferrous iron accumulation, mitochondrial damage, and ER stress. Since NRF2 and SLC7A11 knockdown inhibited HO-1 induction, while HO-1 knockdown did not influence NRF2 and SLC7A11 levels, it seems that the hierarchy axis is NRF2-SLC7A11-HO-1 following BAY treatment [27].

Recently, dimethyl fumarate (DMF), an FDA-approved drug for multiple sclerosis that can protect against inflammation and pro-oxidant conditions, was reported to have a beneficial anti-ferroptosis role in cognitive diseases. These effects are mediated by the joint activities of DMF on the NRF2/ARE and NF-κB signaling pathways [341]. DMF treatment on the one hand decreased NF-κB-mediated pro-inflammatory cytokine expression and on the other hand, it activated NRF2/ARE signaling, which reduces features of ferroptosis, and increased HO-1, NQO1, and GPX4 expression [341]. It has been also described that NF-κB activation in glioblastoma cells by RSL3 treatment leads to an increase of lipid ROS and a reduction of anti-ferroptosis proteins such as GPX4, ATF4, and SLC7A11 [342]. Accordingly, NFκB inhibition with the BAY 11-7082 compound increased the expression of ferroptosis-related proteins, such as ATF4, SLC7A11, and HO-1, consequently lowering RSL3-induced ferroptosis. In addition, murine xenograft models demonstrated that RSL3-treated tumors were approximately one-quarter of the weight compared to control groups, but cotreatment with RSL3 and BAY 11-7082 resulted in a significant increase in tumor weight with respect to the only RSL3-treated tumors. Therefore, the NF-κB pathway could mediate ATF4 and SLC7A11 expression and in the condition of simultaneous GPX4 silencing could induce ferroptosis in glioblastoma cells; in fact, depletion of GPX4 alone is not sufficient to induce ferroptosis of glioblastoma cells and only GPX4 silencing causes a moderate cell proliferation blockage without altering SLC7A11 and ATF4 levels. Conversely, NF-κB pathway activation in GPX4 knocked-down glioma cells decreased ATF4 and SLC7A11 expression and significantly increased the occurrence of ferroptosis [342].

Many studies showed a negative effect of NF-κB on ferroptosis by exhausting cellular iron and preventing lipid peroxidation. Diffuse large B-cell lymphoma (DLBCL), the most frequent malignant lymphoma in adults, is characterized by constitutive NF-κB activation and consequently by a constitutive production of interleukin-6 (IL-6) and IL-10. Dimethyl fumarate (DMF) efficiently inhibited NF-κB signaling (and STAT3) in DLBCL cell lines, which showed a significant sensitivity toward DMF-induced ferroptosis, exhibiting massive lipid peroxidation and reduced GSH levels following the DMF treatment [343]. Gene expression profiling of DMF-treated HBL-1 cells, a human lymphoma cell line, revealed the downmodulation of several NF-κB target genes. Specifically, DMF treatment disrupted IKK complex activation by altering IκBα phosphorylation and consequently NF-κB nuclear translocation. Conversely, overexpression of the NF-κB member RelA decreased ferroptosis induced by DMF treatment, probably through the NF-κB-mediated induction of various antioxidant proteins, such as TRX, NQO1, HO-1, MnSOD, and FTH [343]. Wang et al. demonstrated that in hepatocellular carcinoma cells, NF-κB p65 hindered the in vitro and in vivo ferroptosis induced by aspirin. This effect was mediated by transcriptional activation of SLC7A11 through direct binding of NF-κB p65 to its promoter region. The authors proved that aspirin caused ferroptosis by inhibiting NF-κB p65-activated SLC7A11 transcription, whereas NF-κB p65 overexpression interfered with the aspirin-induced ferroptosis of hepatocellular carcinoma cells [344].

Myrislignan, a kind of lignan that inhibits NF-κB signaling in murine macrophage cells [345], significantly decreases the growth, migration, and invasion abilities of glioblastoma (GBM) cells. Zhou et al. demonstrated that myrislignan treatment in GBM cells inhibits the activation of the NF-κB signaling pathway through a reduction of IκB-α and p65 phosphorylation levels in U87 and U251 glioblastoma cells. By cotreatment with the myrislignan and ferroptosis inhibitor Fer-1, they showed that myrislignan-induced suppression of proliferation is caused by ferroptosis induction. Indeed, myrislignan decreased the levels of cystine and GSH and induced lipid peroxidation in GBM cells without altering iron pool levels. The molecular mechanism of myrislignan-induced ferroptosis involved a decrease in SLC7A11 expression, while NRF2, TFR1, and GPX4 levels remained unchanged. Further, knockdown of Slug/SNAI2, a transcription factor with a key role in TGFB signaling, enhanced the suppression of SLC7A11 level induced by myrislignan, suggesting that myrislignan regulated the ferroptosis of GBM cells via the Slug–SLC7A11 pathway. The effect of myrislignan on GBM was confirmed in vivo in the intracranial xenograft model; in particular, Western blot analysis and IHC staining demonstrated that expression of p-p65, Slug, and SLC7A11 declined in myrislignan-treated tumors. Thus, the authors determined that the NF-κB inhibitor myrislignan drives the ferroptosis of glioblastoma cells through Slug–SLC7A11 signaling [346].

NF-κB signaling could protect from ferroptosis by inducing the expression of lipocalin-2 (LCN2), a cytokine involved in iron homeostasis and inflammation [347]. Recently, Yao et al. reported that leukemia inhibitory factor receptor (LIFR) is frequently downregulated in liver cancer and its loss provokes liver tumorigenesis and confers resistance to drug-induced ferroptosis, as well as to erastin and RSL3 [348]. By RNA seq, the authors also demonstrated that LIFR loss in vivo and in vitro provoked upregulation of the iron-sequestering LCN2 and this was causally associated with induction of NF-κB signaling. Further analyses demonstrated that: (i) LIFR-knockout mouse liver progenitor cell lines (PHM) were resistant to ferroptosis, while LIFR overexpression sensitized PHM cells to ferroptosis inducers; (ii) the knockout of LIFR increased p65 phosphorylation in PHM and human liver cancer cell lines; conversely, LIFR overexpression reduced p65 phosphorylation, which was rescued by the knockdown of phosphatase SHP1; and (iii) knockdown of LIFR increased LCN2 levels, conferring resistance to erastin and sorafenib, which could be reversed by knockdown of LCN2. In conclusion, NF-κB-mediated LCN2 upregulation renders cells resistant to ferroptosis through the LIFR/SHP1 cascade in the liver.

### 4.6. p53 Pathway

The tumor suppressor p53 is a signal-activated sequence-specific DNA-binding protein, consisting of 393 amino acids with two tandem activation domains; a central DNA-binding domain; and a proline-rich domain, crucial for p53 interactions with proteins and for apoptosis regulation; an oligomerization domain necessary for activity; and a basic domain regulating the binding to responsive elements in the promoter regions of target genes [349]. The transcription cofactors CBP/p300 assist p53 in the activation of its target genes, including CDKN1A/p21 and murine double minute-2 (MDM2), through histone acetylation at promoter regions. Also, CBP/p300 can acetylate p53 itself, thus regulating DNA binding transactivation and protein stability [350]. Instead, MDM2, an E3 ubiquitin-protein ligase, is a p53 negative regulator, promoting its rapid degradation under conditions in which p53 is otherwise stabilized [351]. p53 is activated following cellular stress such as hypoxia, carcinogens, and oxidative stress, and coordinates the different cellular responses associated with DNA repair, apoptosis, ferroptosis, and energy metabolism. In particular, its activation leads to cell-cycle arrest favoring DNA repair or cell death induction by diverse pathways [66,352].

In recent years, an increasing number of works have highlighted that p53 is a critical factor in the regulation of ferroptosis. In particular, p53 has a dual role as it can promote or prevent ferroptosis, depending on the cell/tissue context and also on the type and the severity of stress, as well as damage upon stress exposure [66]. p53 activities in ferroptosis occur through transcriptional regulation of downstream genes and/or through diverse post-transcriptional modifications (PTMs) by which p53 activity are also regulated. p53 PTMs can be rapidly reversed, thus regulating its activity and function, and this represents a crucial event that markedly influences ferroptosis. For example, suppressor of cytokine signaling 1 (SOCS1) mediates p53 phosphorylation at Ser15, leading to ferroptosis-related downregulation of SLC7A11 and upregulation of spermidine/spermine N1-acetyltransferase 1 (SAT1) [352,353], while the attenuation of p53 phosphorylation at Ser46 in hepatocellular carcinoma HepG2, mediated by zinc finger protein 498 (ZNF498), lowered ROS production and GSL2 expression, and raised GSH production, with a consequent increase of cell survival and p53-mediated ferroptosis inhibition [354]. Also, CBP acetylates p53 (at residue K98 in the mouse and residue K101 in the human) without affecting the p53 DNA-binding capacity or transcriptional activity and this PTM retained p53’s ability to repress SLC7A11 expression and induce ferroptosis Instead, acetylation of p53 at K320 activates CDKN1A/p21 expression, hindering ferroptosis [64].

p53 also regulates several fundamental metabolic pathways involved in ferroptosis, providing control of GSH, glutamine, iron, and lipid peroxide levels. The molecular mechanism by which p53 mediates the metabolic regulation of ferroptosis was firstly reported by Le Jiang et al. [355], who identified SLC7A11 as a novel p53 target gene. The authors produced a tetracycline-controlled (tet-on) p53-inducible human non-small cell lung carcinoma H1299 cell line for microarray analysis, demonstrating that SLC7A11 mRNA and protein expression levels were strongly reduced upon p53 activation. On the contrary, SLC7A11 up-regulation was observed in p53-knockdown cells. p53 directly binds to a consensus sequence in the 5′ flanking region of the SLC7A11 gene and similar results were reproduced in other human cancer cell lines with wild-type p53 (H460 and MCF-7), while no effects were observed in p53-null cells (H1299 and SAOS-2). SLC7A11 overexpression counteracted erastin-induced ferroptosis and tumor growth suppression induced by p53 in xenograft tumor models. Also, mouse embryo fibroblasts (MEFs) derived from BAC transgenic mice overexpressing SLC7A11 (Slc7a11-BAC MEFs) treated with either ROS or erastin demonstrated that high levels of SLC7A11 reduced ferroptosis in wild-type MEFs. Therefore, p53 activation by decreasing SLC7A11-mediated cystine uptake can, in turn, limit the intracellular GSH concentration and increase ferroptosis cell death in the presence of ROS stress. Accordingly, p53 induced by nutlin-3 or by exposure to DNA damage stress (stimulating p53 function) sensitizes the cancer cells to ferroptosis [355].

p53 regulates iron levels through ferredoxin reductase (FDXR) and solute carrier family 25 member 28 (SLC25A28), which are two targets of p53, located in mitochondria. In the first case, Zhang et al. generated a Fdxr-deficient mouse model and found that Fdxr deficiency leads to embryonic lethality associated with iron overload with increased expression of IRP2 that in turn could block p53 mRNA translation, implying that FDXR may play a role in p53-dependent ferroptosis. That study proposed a model in which this FDXR–IRP2–p53 loop impacts iron metabolism [356]. Another paper reported that p53 regulates mitochondrial iron homeostasis in hepatic stellate cells by enhancing the activity of SLC25A28, which is involved in iron trafficking into mitochondria [357] The molecular mechanism involves bromodomain-containing protein 7 (BRD7)-mediated mitochondrial translocation of p53, which in turn directly interacts with SLC25A28, provoking iron overload and ETC hyperfunction, which consequently increases sensitivity to ferroptosis [357]. Another metabolic gene regulated by p53 and promoting ferroptosis is spermidine/spermine N1-acetyltransferase 1 (SAT1), a key enzyme of polyamine catabolism that was induced by nutlin-3 or doxorubicin treatment in cancer cell lines expressing wild-type p53 (U2OS, MCF7, and A375) but not in the p53-null cell line H1299 [358]. SAT1 transcription is directly upregulated by p53, which binds two sites on its promoter region. The injection of the SAT1 Tet-on H1299 cells into nude mice and induction of the SAT1 expression in xenograft mice showed a dramatic reduction of tumor growth. Treatment of SAT1 Tet-on cells with the ROS-inducing agent tert-butyl hydroperoxide (TBH) revealed that the combination of SAT1 over-expression and ROS treatment induced significant ferroptosis accompanied by up-regulation of prostaglandin-endoperoxide synthase 2 (COX2), identified as a potential molecular marker of ferroptosis [359], and an elevated lipid ROS level. Moreover, p53 activation by nutlin and ROS stress failed to induce ferroptosis in SAT1-knockout cells, indicating that SAT1 contributes to p53-mediated ferroptosis [358]. The authors identified the downstream effector of p53-induced SAT1 in the ALOX15 enzyme, firstly found in HT1080 [148]. Thus, p53 indirectly, by modulating the expression of ALOX15 via SAT1, affects phospholipid peroxidation in PUFA and initiates ferroptosis by an independent GSH and GPX4 axis [360]. Small molecules with the potential to be ferroptosis inducers in cancer cells are analogs sharing a common 4-cyclopentenyl-2-ethynylthiazole skeleton called CETZOLES [361]. Among these, CETZOLE1 blocks cystine uptake, leading to GSH loss, elevation of ROS, and lipid peroxidation [362]. Indeed, Nishanth Kuganesan et al. showed that p53^−^/^−^ MEFs were less sensitive to CETZOLE 1 and to RSL3 than wild-type MEFs. Also, inducible systems in human adult fibroblasts TR9-7 cells for p53 levels displayed enhanced ferroptosis under CETZOLE1 treatment, and liproxstatin-1 decreased TR9-7 cell death provoked by CETZOLE1. Besides, the increased p53 in these cells caused a significantly elevation of membrane lipid peroxidation induced by CETZOLE 1 treatment. The authors also demonstrated that p21 can suppress CETEZOLE1-induced ferroptosis; therefore, p53 mediated ferroptosis in a p21-independent manner [362]. Indeed, CETZOLE1 does not induce ferroptosis in cells over-expressing p21 [362,363,364].

In some studies, p53 was proposed to inhibit ferroptosis in certain cell types, including HT1080 fibrosarcoma cells, that express wild-type p53. Stabilization of wild-type p53 by pretreatment of these cells with nutlin-3 makes them less responsive to system Xc- inhibition by erastin2 and this lower sensitivity requires p21 and preservation of the GSH content [365]. A similar effect was observed in the human osteosarcoma SJSa1 cell line, but not in human fetal lung fibroblasts IMR-90, where p53 stabilization does not affect erastin2-induced ferroptosis. Conversely, lung adenocarcinoma cell lines derived from p53 transactivation function-deficient mutant mice [366] do not suppress ferroptosis induced by erastin2 (an erastin analog), suggesting that p53 delays ferroptosis via transcriptional activation of its target genes, including p21 [365]. Of note, previous experiments have demonstrated that p21 could increase NRF2 levels by competing with Keap1-mediated NRF2 ubiquitination [367]. Therefore, p53 activation may also induce NRF2 activity via p21 increase to regulate an antioxidant gene expression program that preserves intracellular GSH [368].

p53 protective functions in ferroptosis can also be also independent of its transcriptional activity, as reported by Xie et al. in human colorectal cancer (CRC) cells [105]. In that paper, the sensitivity to erastin of wild-type p53 in CRC cells was restored following p53 silencing by transfection with small hairpin RNAs or by knockdown/pharmacological inhibition of p53, accompanied by an increase in erastin-induced SLC7A11 mRNA and protein expression levels. The authors found that p53 also inhibits erastin-induced ferroptosis by blocking dipeptidyl-peptidase-4 (DPP4) protease activity in a transcription-independent manner. In detail, DPP4 knockdown abrogates erastin-induced death of p53-knockdown CRC cells, indicating that DPP4 inactivation is required for ferroptosis in p53-deficient CRC cells. p53 downmodulation and real-time qPCR showed that p53 does not alter the mRNA expression level of DPP4; in the meantime, immunoblots and enzymatic activity assays revealed that p53 influences the subcellular localization and activity of DDP4. DDP4 can function as a serine protease at the plasma membrane, where it forms a complex with NOX1, contributing to lipid peroxidation and iron accumulation, and as a transcription cofactor in the nucleus. Thus, p53 favors the subcellular redistribution of DPP4 toward a nuclear enzymatically inactive pool. In conclusion, p53 limits ferroptosis by blocking DPP4 activity [105].

It is worth noting that many of the above TFs participating in the ferroptosis process can have common target genes, in some cases with opposite effects (Table 1), thus conceptualizing transcriptional pathways to and enabling a complex gene regulatory network impacting the redox system and the sensitivity profiling of ferroptosis.

## 5. Conclusions

Ferroptosis is a newly discovered distinct type of regulated cell death associated with dysregulated iron homeostasis, ROS accumulation, and lipid peroxidation. Since its discovery in 2012, a large body of evidence has been gathered on the molecular mechanisms of ferroptosis and great efforts have been made to identify the effectors and potential inducers and inhibitors of this process. Furthermore, a link between ferroptosis and a variety of pathological processes and aging-related disorders has rapidly emerged, thus attracting extensive research interest in the potential therapeutic applications of ferroptosis induction or inhibition in several diseases. Nevertheless, research in this field is still in its early stages, and several questions need to be answered before considering ferroptotic-based treatments as an effective therapeutic option: (1) understanding the physiological functions of ferroptosis is still a challenging issue and needs to be clarified to evaluate possible preventive treatments, particularly in degenerative disorders; (2) it is evident that ferroptosis is a tissue- and cell-type dependent process; therefore, not all cancer types or even different clinical stages of the same type of cancer can be treated with the same ferroptosis-based cancer therapies; (3) further studies are required to clarify possible cross-talks between ferroptosis and other regulated cell death pathways that could contribute to more efficiently circumventing drug resistance in cancer.

Bearing this in mind, it is evident that a better understanding of the molecular mechanisms and the metabolic pathways controlling ferroptosis is of relevant importance. Given the broad function of transcription factors in antioxidant circuits, we envision that reprogramming of the intracellular antioxidant state needs to be considered as a novel therapeutic approach to the many human diseases that have ferroptosis as a common trait.

## Figures and Tables

**Figure 1 antioxidants-13-00298-f001:**
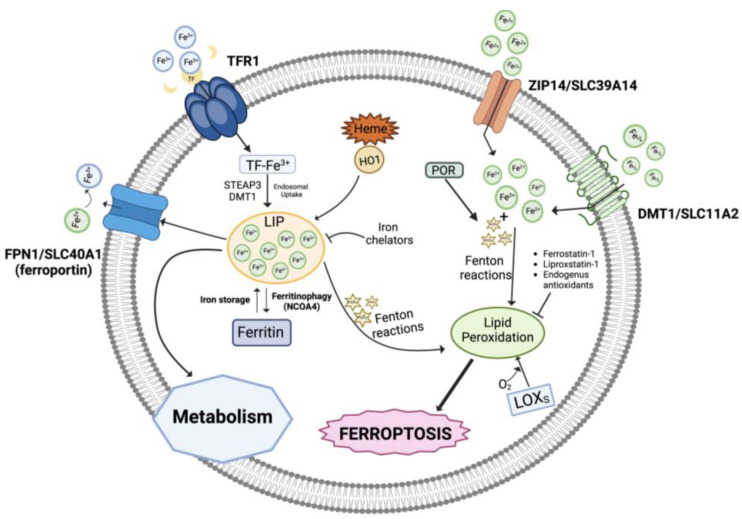
Role of iron in ferroptosis. An overview of the iron metabolism showing the transport mechanisms, storage, and intracellular metabolism. LIP, labile iron pool; TFR1, transferrin receptor 1; HO-1, heme oxygenase 1; POR, NADPH-cytochrome P450 reductase; LOXs, lipoxygenases. Created with BioRender.com (accessed on 31 January 2024).

**Figure 2 antioxidants-13-00298-f002:**
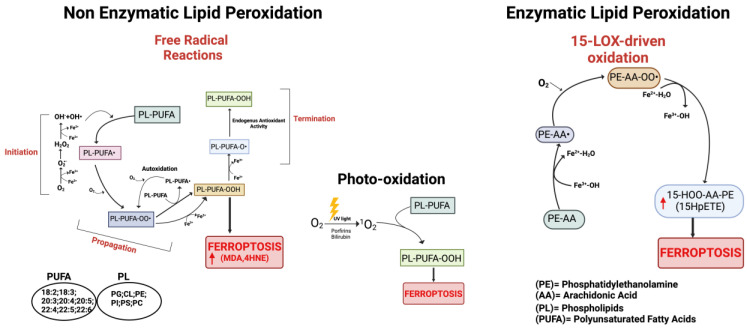
Schematic representation of the three main pathways involved in the process of lipid peroxidation. On the left: non-enzymatic lipid peroxidation producing lipid peroxyl radicals (PL-PUFA-OO^•^). In the center: lipid peroxidation mediated by non-radical singlet oxygen (^1^O_2_). On the right: enzymatic lipid peroxidation cascade; the mechanism of 15-Lipoxygenase is represented. Red arrows indicate increased products. Created with BioRender.com (accessed on 31 January 2024).

**Figure 3 antioxidants-13-00298-f003:**
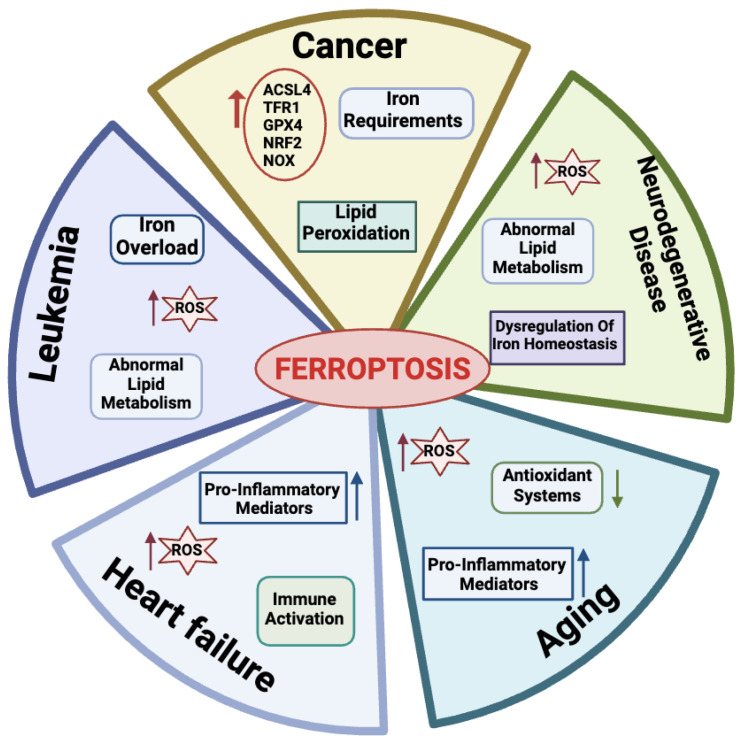
Physiological and pathological processes related to ferroptosis. For each disease, the main features are reported. Created with BioRender.com (accessed on 30 January 2024).

**Figure 4 antioxidants-13-00298-f004:**
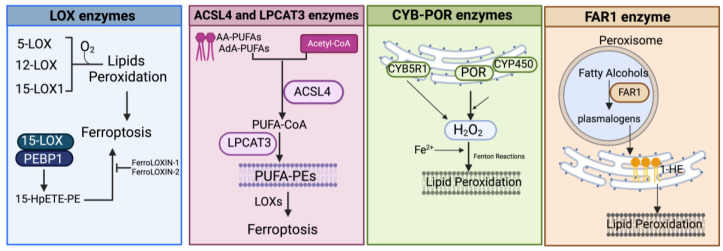
Schematic representation of the main enzymatic pathways mediating the process of lipid peroxidation in ferroptosis. LOX, lipoxygenase; PEBP1, phosphatidylethanolamine-binding protein; 15-HpETE-PE, 15-hydroperoxyphosphatidylethanolamines; AA-PUFAs, PUFA arachidonic acid; AdA-PUFAs, adrenic acid; PUFA-PEs, PUFA-phosphatidylethanolamines; CYB5R1, NADH-cytochrome b5 reductase 1; POR, NADPH-cytochrome P450 reductase; CYP, cytochrome P450s; FAR1, fatty acyl-CoA reductase1; 1-HE, 1-hexadecanol. Created with BioRender.com (accessed on 30 January 2024).

**Figure 5 antioxidants-13-00298-f005:**
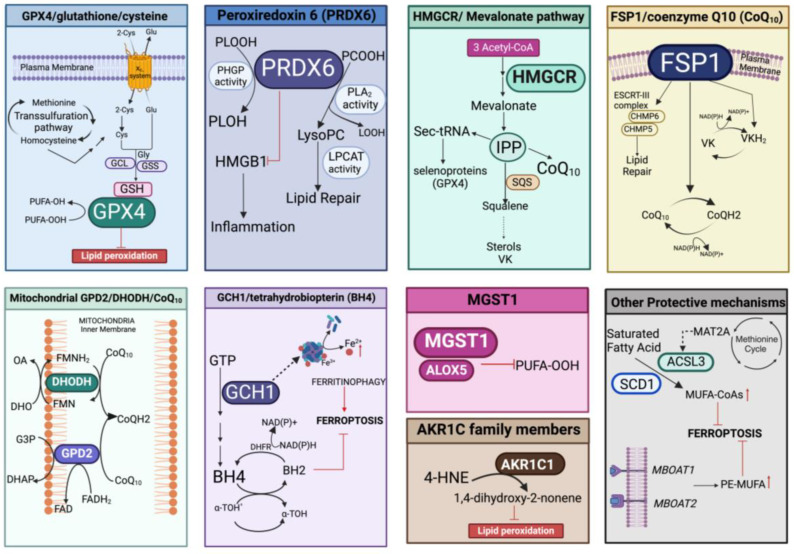
Overview of the different enzymatic pathways involved in prevention/protection against lipid peroxidation. In each panel, the specific enzymatic cascade and cofactors evoking inhibition of ferroptosis are represented. PLOOH, peroxidized phospholipids; PLOH, lipid alcohol; GCL, glutamate cysteine ligase; GSS, glutathione synthase; PCOOH, phosphatidylcholine peroxide; LOOH, hydroperoxyl lipid; lysoPC, lysophosphatidylcholine; LPCAT, lysophosphatidylcholine acyltransferase 3; IPP, isopentenyl pyrophosphate; SQS, squalene synthase; Sec-tRNA, selenocysteine (Sec) tRNA; ESCRT-III, endosomal sorting complexes required for transport III; CHMP5/6, charged multivesicular body protein 5/6; VK, vitamin K; VKH2, vitamin K hydroquinone; GPD2, the glycerol-3-phosphate dehydrogenase 2; DHODH, the dihydroorotate dehydrogenase; G3P, glycerol-3-phosphate; DHO, dihydroorotic acid; OA, orotic acid; DHAP, dihydroxyacetone phosphate; FAD, flavin dinucleotide; FADH2 reduced flavin dinucleotide; FMN, flavin mononucleotide; FMNH2, reduced flavin mononucleotide; GCH1, GTP cyclohydrolase 1; BH_4_, tetrahydrobiopterin; BH2, dihydrobiopterin; α-TOH• α-tocopheroxyl radical; a-TOH, α-tocopheroxyl; DHFR, dihydrofolate reductase; MGST1, microsomal glutathione transferase 1; ALOX5, arachidonate 5-lipoxygenase; AKR1C1, aldo-keto reductase family 1 member C14-HNE, 4-hydroxynonenal; MAT2A, methionine adenosyltransferase 2α; ACSL3, acyl-CoA synthetase long chain family member 3; SCD1, stearoyl-CoA desaturase 1; MUFA-CoAs, Monounsaturated fatty acyl-CoAs; PE-MUFA, phosphatidylethanolamine monounsaturated fatty acid. Created with BioRender.com (accessed on 31 January 2024).

**Figure 6 antioxidants-13-00298-f006:**
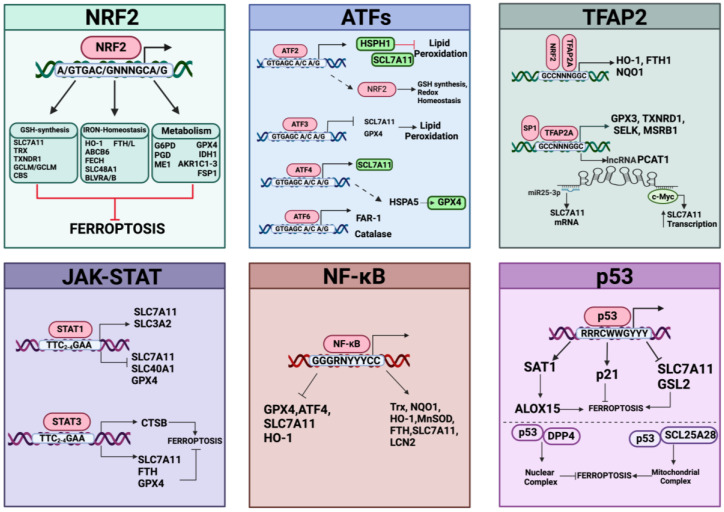
Transcription factors and their redox-related gene targets in ferroptosis. For each transcription factor, DNA regulatory elements, target genes, and their positive or negative effects on ferroptosis are shown. Created with BioRender.com (accessed on 30 January 2024).

**Table 1 antioxidants-13-00298-t001:** Common target genes regulated by different transcription factors. The green arrow indicates up-regulation and the red arrow indicates down-regulation. The presence of both arrows indicates that both effects have been demonstrated, depending on the different tissue and cellular contexts.

Direct Target Genes	Transcription Factors and Effects on Target Genes	
*FTH*	NRF2	⭡
	TFAP2	⭣
	STAT3	⭡
*GPX4*	NRF2	⭡ ⭣
	ATF3	⭣
	STAT1	⭣
	STAT3	⭡
	NF-kB	⭡
*HO1*	NRF2	⭡
	TFAP2	⭣
	NF-kB	⭡
*NQO1*	NRF2	⭡
	TFAP2	⭡
	NF-kB	⭡
*SCL7A11*	NRF2	⭡
	ATF4	⭡
	ATF3	⭣
	STAT1	⭡ ⭣
	STAT3	⭡
	NF-kB	⭡ ⭣
	P53	⭣
*TXNDR1*	NRF2	⭡
	TFAP2	⭡

## Data Availability

No new data were created or analyzed in this study. Data sharing is not applicable to this article.

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
