# Peer review of "Antioxidant Systems as Modulators of Ferroptosis: Focus on Transcription Factors"

_antioxidants, 2024, doi:10.3390/antiox13030298_

Round 1
Reviewer 1 Report
1. In Fig.3, there were leukemia and heart failure, which weren’t discussed in the review.
2. The full name needs to be shown when an abbreviation appears in the review for the first time.
Do the transcription factors participating in the ferroptosis have common target genes? If yes, how are the common target genes regulated by different transcription factors?
Author Response
Manuscript ID: antioxidants-2876370
Dear Editor,
all the authors are very grateful to the reviewers for the positive comments and for their suggestions.
We have critically reviewed their comments and revised the manuscript as per reviewers’ suggestions. Our point by point responses to their comments are addressed in color below, and the revised portion has been indicated in red color in the revised manuscript. In addition, we have included a Table (Table1) to show common target genes that are regulated by different transcription factors (see comment of Reviewer 1)
Responses to Reviewer #1 Major comments
1. In Fig.3, there were leukemia and heart failure, which weren’t discussed in the review.
Response: We thank the Reviewer for this professional comment. According to this, we have added in the revised version, two additional sub-paragraphs (2.4.3 and 2.4.4) on leukemia and heart failure, respectively.
2. The full name needs to be shown when an abbreviation appears in the review for the first time.
Response: We thank the Reviewer for this comment. The abbreviations have been carefully revised and the full name used only for the first time.
Detail comments
Do the transcription factors participating in the ferroptosis have common target genes? If yes, how are the common target genes regulated by different transcription factors?
Response: We thank the Reviewer for the insightful comments. We acknowledge that numerous transcription factors participating in the ferroptosis have common target genes, in some cases producing opposite effects. Therefore, we present a Table (Table 1) in which we showed common targets genes regulated by different transcription factors with their related effect/s. We believe that this schematic representation will help the reader to follow the intricate transcriptional network of ferroptosis. The relative part is included in the manuscript at the end of paragraph 4.

Reviewer 2 Report
In this review, Punziano et al. give an extensive overview and comprehensive summarization of the rather young cell death pathway called Ferroptosis. Such a critical review, which sheds light on every parameter in the very complicated interplay between ROS sources, detoxification mechanisms, cellular outcomes, oxidation products (especially inside and outside of membranes) and the involvement in diseases was in high need. In general, this review is one of the best I have red in as reviewer I “Antioxidants”. Overall the writing, phrasing and grammar of the manuscript are excellent and understandable. Very well done. All of the topics are sufficiently explained. I have only a few minor suggestions to further improve this already impressive article.
Minor points for the text:
In line 28: Please replace “potential” with “balance”
In line 31: I think the authors mean autophagy-induced cell death. I read this very often. Autophagy is not a cell death pathway per se, meaning it is not its sole function. While necroptosis, apoptosis, ferroptosis all leading to one final outcome i.e. cell death, autophagy is basically a recycling pathway. However, cell death occurs, when this pathway is disturbed. Then the term “autophagy-induced cell death” is used, which, in my opinion, is also misleading. Perhaps the authors can just differ at least between autophagy and autophagy-induced cell death with one or two sentences.
In line 23: Please remove “is” after ferroptosis.
In lines 26-27: In this sentence “as well as” is used two times, which always sounds funny. Please rephrase.
In line 48: The abbreviations for ROS and RNS were introduced earlier in the text already.
In Figure 1: The H2O2 molecules seem a little of place. It somehow looks like they are kind of separated from the Fenton reaction or, even worse, a result of it. You could place them near to the arrow curve, which emerges from LIP and goes to Lipid oxidation and also place them near to the vertical arrow reaching from the free Fe2+ to Lipid oxidation. I think it should be more clear that H2O2 is consumed during the Fenton reactions.
In line 335: The authors nicely summarize and discuss Nox enzymes and their ROS subspecies output in lines 355-369. Therefore, the text passage about Nox4 and the Duox enzymes as sources for H2O2 is anticipatory and also here not necessary. As correctly stated later by the authors, all Nox enzymes generate O2- as first product, because the electrons are always transferred to oxygen. In all occasions, O2- is rapidly transmutated to H2O2 by various ways already described in the text. However, Nox4 and the Duox enzymes (because of their structure) just retain O2- longer, which allows is spontaneous dismutation to H2O2, which is then released from these special Nox isoforms. I am happy that the authors did not incorporate the “peroxidase domain” of the Duox enzymes in the text. Researchers always write that the peroxidase domain generates H2O2, which is enzymatically wrong. The peroxidase domain consumes the H2O2. However, this was only proven in Drosophila melanogaster so far.
If these points can be addressed by the authors in a minor revision, this already impressive review is ready for publication. Good job.
Author Response
Manuscript ID: antioxidants-2876370
Dear Editor,
all the authors are very grateful to the reviewers for the positive comments and for their suggestions.
We have critically reviewed their comments and revised the manuscript as per reviewers’ suggestions. Our point by point responses to their comments are addressed in color below, and the revised portion has been indicated in red color in the revised manuscript. In addition, we have included a Table (Table1) to show common target genes that are regulated by different transcription factors (see comment of Reviewer 1)
Responses to Reviewer #2 Major comments
In this review, Punziano et al. give an extensive overview and comprehensive summarization of the rather young cell death pathway called Ferroptosis. Such a critical review, which sheds light on every parameter in the very complicated interplay between ROS sources, detoxification mechanisms, cellular outcomes, oxidation products (especially inside and outside of membranes) and the involvement in diseases was in high need. In general, this review is one of the best I have red in as reviewer I “Antioxidants”. Overall the writing, phrasing and grammar of the manuscript are excellent and understandable. Very well done. All of the topics are sufficiently explained. I have only a few minor suggestions to further improve this already impressive article.
Response: We greatly appreciate the very nice comments of the Reviewer that help us to be confident in our manuscript.
Detail comments
Minor points for the text:
In line 28: Please replace “potential” with “balance”
Response: We thank the Reviewer for this suggestion. The word was replaced.
In line 31: I think the authors mean autophagy-induced cell death. I read this very often. Autophagy is not a cell death pathway per se, meaning it is not its sole function. While necroptosis, apoptosis, ferroptosis all leading to one final outcome i.e. cell death, autophagy is basically a recycling pathway. However, cell death occurs, when this pathway is disturbed. Then the term “autophagy-induced cell death” is used, which, in my opinion, is also misleading. Perhaps the authors can just differ at least between autophagy and autophagy-induced cell death with one or two sentences.
Response: We thank the Reviewer for this professional comments. According to the above suggestion we have added in the revised version, a sentence to distinguish autophagy from autophagy-induced cell death.
In line 23: Please remove “is” after ferroptosis.
Response: We thank the Reviewer for this suggestion. The word was removed.
In lines 26-27: In this sentence “as well as” is used two times, which always sounds funny. Please rephrase.
Response: We thank the Reviewer for this suggestion. We rephrase the sentence.
In line 48: The abbreviations for ROS and RNS were introduced earlier in the text already.
Response: We thank the Reviewer for this suggestion. The words ROS and RNS were removed.
In Figure 1: The H2O2 molecules seem a little of place. It somehow looks like they are kind of separated from the Fenton reaction or, even worse, a result of it. You could place them near to the arrow curve, which emerges from LIP and goes to Lipid oxidation and also place them near to the vertical arrow reaching from the free Fe2+ to Lipid oxidation. I think it should be more clear that H2O2 is consumed during the Fenton reactions.
Response: We thank the Reviewer for this good suggestion. Accordingly, the H2O2 molecules were placed near to the arrow curve, which emerges from LIP and goes to Lipid oxidation, and placed near to the vertical arrow reaching from the free Fe2+ to Lipid oxidation.
In line 335: The authors nicely summarize and discuss Nox enzymes and their ROS subspecies output in lines 355-369. Therefore, the text passage about Nox4 and the Duox enzymes as sources for H2O2 is anticipatory and also here not necessary. As correctly stated later by the authors, all Nox enzymes generate O2- as first product, because the electrons are always transferred to oxygen. In all occasions, O2- is rapidly transmutated to H2O2 by various ways already described in the text. However, Nox4 and the Duox enzymes (because of their structure) just retain O2- longer, which allows is spontaneous dismutation to H2O2, which is then released from these special Nox isoforms. I am happy that the authors did not incorporate the “peroxidase domain” of the Duox enzymes in the text. Researchers always write that the peroxidase domain generates H2O2, which is enzymatically wrong. The peroxidase domain consumes the H2O2. However, this was only proven in Drosophila melanogaster so far.
Response: We thank the Reviewer for the careful observations on this part of the manuscript. Accordingly, we eliminate the anticipation about Nox4 and Duox enzymes of line 335. Furthermore, we revised the original lines 355-369 (now lines 401-402) highlighting that Nox4 and the Duox enzymes retain O2- longer, which allows is spontaneous dismutation to H2O2, which is then released from these special Nox isoforms.
If these points can be addressed by the authors in a minor revision, this already impressive review is ready for publication. Good job.
Response: We thank the Reviewer for considering our work to be of quality.
